# ADVERSARIAL ROBUSTNESS OF SELF-SUPERVISED LEARNING IN VISION

## ABSTRACT

Self-supervised learning (SSL) has advanced significantly in visual representation learning, yet large-scale evaluations of its adversarial robustness remain limited. In this study, we evaluate the adversarial robustness of seven SSL models and one supervised model across a range of tasks, including ImageNet classification, transfer learning, segmentation, and detection. Our findings demonstrate that SSL models generally exhibit superior robustness to adversarial attacks compared to their supervised counterpart on ImageNet, with this advantage extending to transfer learning in classification tasks. However, this robustness is less pronounced in segmentation and detection tasks. We also explore the role of architectural choices in model robustness, observing that their impact varies depending on the SSL objective. Finally, we assess the effect of extended training durations on adversarial robustness, finding that longer training may offer slight improvements without compromising robustness. Our analysis highlights promising directions for enhancing the adversarial robustness of visual self-supervised representation systems in complex environments.

## 1 INTRODUCTION

Self-supervised learning (SSL) Balestriero et al. (2023) has emerged as a foundational approach for training models with remarkable capabilities in areas such as language Touvron et al. (2023), vision Oquab et al. (2024), and decision-making Kim et al. (2024). As these models become increasingly widespread and integrated into various applications, ensuring their reliability and safety has become a critical concern Bommasani et al. (2022); Bengio et al. (2024).

One particular challenge is the surprising vulnerability of deep learning models to adversarial examples, where slight input alterations can significantly impact model performance Szegedy et al. (2013); Goodfellow et al. (2014). This phenomenon has sparked significant debate, seeking to understand and mitigate these vulnerabilities Fawzi et al. (2016); Tanay & Griffin (2016); Shafahi et al. (2020); Schmidt et al. (2018); Wang et al. (2022; 2020); Wu et al. (2020); Bai et al. (2022). One prominent theory Ilyas et al. (2019) suggests that adversarial examples arise from the model's sensitivity to non-robust features in the input data. According to this view, both robust (stable) and non-robust (vulnerable) features contribute to classification, with adversarial attacks manipulating the latter to cause misclassification. However, this theory, developed primarily in the context of supervised learning, faces challenges when extended to other self-supervised paradigms. Li et al. (2024) indicates that non-robust features are less effective in SSL methods such as contrastive learning Chen et al. (2020b), masked image modeling He et al. (2021), or diffusion models Ho et al. (2020). This discrepancy suggests that non-robust features may lack the transferability across learning paradigms that robust or natural features possess. Thus, it becomes essential to investigate the model once more, particularly in contexts like SSL, where there is a need for comprehensive research on the adversarial robustness of SSL models.

Notwithstanding the progress made in understanding the adversarial robustness of SSL, particularly contrastive learning, which we extensively discuss in section 2, several key questions remain unresolved. First, with the wide variety of self-supervised representations available, employing different pretext tasks and data augmentations, which approaches demonstrate the greatest adversarial robustness? This remains unclear since most methods don't provide any results on adversarial robustness unless it is a specific focus of the proposed approach. Secondly, robustness is typically assessed

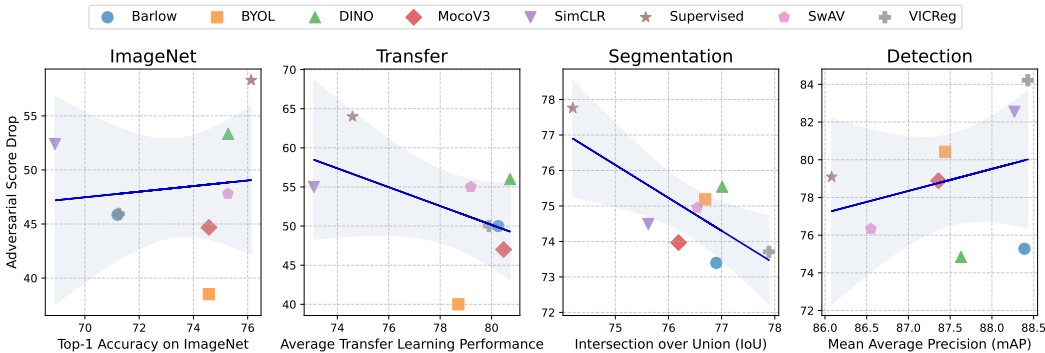

Figure 1: Performance scores for tasks such as ImageNet classification, transfer learning, segmentation, and detection, are shown in relation to the percentage drop in adversarial robustness. The shaded regions indicate the 95% confidence interval around the regression line.

by the model's accuracy on the pretraining dataset. Still, its adversarial impact on other object recognition datasets or downstream tasks like detection and segmentation has not been thoroughly investigated Kowalczuk et al. (2024).

The choice of model architecture also raises questions about robustness. Standard vision SSL pre-training typically utilizes a ResNet He et al. (2015) as the backbone, but more recently, larger and more powerful models Chen* et al. (2021); Caron et al. (2021); Oquab et al. (2024) have been developed using vision transformers Dosovitskiy et al. (2021). This leads to the question: Which architecture demonstrates greater robustness under the same SSL objective and with comparable parameter sizes?

Another factor to consider is the training duration. State-of-the-art SSL models are trained for longer durations compared to their supervised counterparts. Several studies indicate that this extended training consistently enhances performance, raising the question of whether this might compromise the models' adversarial robustness.

To address these questions and others, we carry out an extensive empirical benchmarking study on the adversarial robustness of various pre-trained SSL models. Specifically, we assess seven different SSL models, namely Barlow Twins Zbontar et al. (2021), BYOL Grill et al. (2020), DINO Caron et al. (2021), MoCoV3 Chen* et al. (2021), SimCLR Chen et al. (2020b), SwAV Caron et al. (2020), and VICReg Bardes et al. (2022), alongside a supervised model against over 20 distinct IAA (Instance Adversarial Attacks) Chakraborty et al. (2018) and UAP (Universal Adversarial Perturbations) Chaubey et al. (2020) on ImageNet Russakovsky et al. (2015) and nine other image recognition datasets Maji et al. (2013); Fei-Fei et al. (2004); Krause et al. (2013); Krizhevsky (2009); Cimpoi et al. (2013); Nilsback & Zisserman (2008); Bossard et al. (2014); Parkhi et al. (2012). Furthermore, we evaluate their adversarial robustness in segmentation Everingham et al. and detection Dalal & Triggs (2005) tasks, with over five attacks to each. To guide our investigation, we address the key questions outlined below, aiming to provide a comprehensive understanding of adversarial robustness in SSL models.

1. **How does the adversarial robustness of various SSL models compare to that of supervised models on the ImageNet?**
   We find that all SSL models demonstrate greater robustness than the supervised model, both in terms of final performance and the drop in adversarial accuracy. Our results contrast with the previous study Gupta et al. (2022) that suggests contrastive learning, particularly SimCLR, lags behind supervised learning. While this holds true when considering only Instance Adversarial Attacks (IAA), including Universal Adversarial Perturbations (UAP) reveals that the supervised model performs exceptionally poorly. Notably, MoCoV3 exhibits the highest robustness under IAA, despite using a contrastive objective. Furthermore, non-contrastive methods generally outperform SimCLR and supervised learning, except DINO under IAA, though all SSL models perform well against UAP. Our findings highlight that

SSL models are indeed more robust than supervised ones, but the diversity of attacks is crucial in assessing adversarial robustness.

2. **Can SSL models retain robustness in downstream tasks like transfer learning, segmentation, and detection?**
   While our robustness findings on ImageNet generalize to transfer learning in classification, where SSL models not only show robustness but also significantly outperform supervised models, we find that in segmentation and detection tasks, the models exhibit very similar performance and robustness and do not reflect ImageNet results.

3. **What architectures showcase better robustness under the same SSL objective and comparable parameter sizes?**
   Interestingly, we observe that MoCoV3 shows reduced robustness with vision transformers, whereas DINO's robustness improves significantly, bringing it in line with other top-performing SSL models when using ResNet which demonstrates that neither excels over the other and significantly influenced by the SSL objective.

4. **Does longer training in SSL models lead to weakening adversarial robustness?**
   We evaluate SwAV and MoCoV3, each with several checkpoints trained for different numbers of epochs, and find that training longer does not reduce adversarial performance; in fact, it slightly enhances it in both cases.

## 2 RELATED WORK

**Self Supervised Learning** Self-supervised learning(SSL) seeks to extract meaningful and general representations from unlabeled data by leveraging pretext tasks. These tasks can vary, such as predicting the next word Radford & Narasimhan (2018) or neighboring words Devlin et al. (2019) in a text, reconstructing masked sections of an image He et al. (2021), or ensuring that two different perspectives of the same image result in similar visual representations Chen et al. (2020b).

Avoiding collapse is a key challenge in SSL for computer vision, and various methods can be classified based on how they address this issue. Contrastive approaches like SimCLR Chen et al. (2020b) and MoCo He et al. (2019); Chen et al. (2020c); Chen* et al. (2021) use an objective that pushes apart representations of different inputs (negative samples) while bringing together those of the same input (positive samples). The performance and scalability of these methods heavily depend on the number and selection of negative samples. In another category, distillation methods such as BYOL Grill et al. (2020), SimSiam Chen & He (2020), and DINO Caron et al. (2021), prevent collapse by introducing asymmetry between different encoder branches and employing algorithmic adjustments [26]. Additional SSL techniques, including DeepCluster Caron et al. (2019), SeLa Asano et al. (2020), and SwAV Caron et al. (2020), enforce a clustering structure in the feature space to avoid constant representations. Meanwhile, methods like Barlow Twins Zbontar et al. (2021), Whitening MSE (W-MSE) Ermolov et al. (2021), VICReg Bardes et al. (2022), CorInfoMax Ozsoy et al. (2022) prevent collapse by using feature decorrelation.

**Adversarial Self-Supervised Learning** While self-supervised learning (SSL) has outperformed supervised training Chen et al. (2020b), numerous studies highlight that contrastive learning remains susceptible to adversarial attacks when transferring the learned features to downstream classification tasks Ho & Vasconcelos (2020); Kim et al. (2020). To improve the robustness of contrastive learning, adversarial training has been adapted to self-supervised settings. In the absence of labels, adversarial examples are generated by maximizing the contrastive loss with respect to all input samples. Several prior works, such as ACL Jiang et al. (2020), RoCL Kim et al. (2020), and CLAE Ho & Vasconcelos (2020), adopt this approach. Additionally, ACL incorporates the dual-BN technique Xie et al. (2020) to further enhance performance. DeACL Zhang et al. (2022) introduces a two-stage approach, distilling a standard pretrained encoder through adversarial training. Nguyen et al. (2022) establishes an upper bound on the adversarial loss of a prediction model, which is based on the learned representations, for any downstream task. This upper bound is determined using the model's loss on clean data and a robustness regularization term, which helps make the prediction model more resistant to adversarial attacks. Gupta et al. (2022) demonstrates that adversarial sensitivity stems from the uniform distribution of data representations on a unit hypersphere in the representation space. The presence of false negative pairs during training contributes to this effect, increasing the model's vulnerability to input perturbations.

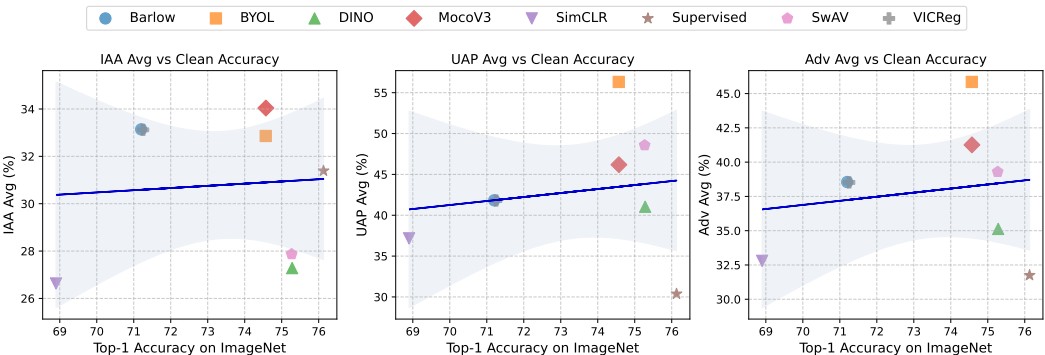

Figure 2: Averaged scores of SSL models on ImageNet across various attack types, including Instance Adversarial Attacks (IAA) and Universal Adversarial Perturbations (UAP). *Adv Avg* refers to the average score across all attacks combined. The shaded regions indicate the 95% confidence interval around the regression line.

Although self-supervised adversarial training has made progress, it still does not match the performance of supervised methods. Luo et al. (2023) suggest that this shortfall is due to data augmentation and propose a dynamic data augmentation scheduler to achieve comparable results to supervised training. Xu et al. (2023) efficiently apply ACL on the ImageNet Russakovsky et al. (2015) to obtain a robust representation using robustness-aware core set selection.

**Robustness of Self-Supervised Learning**

Hendrycks et al. (2019) found that incorporating an extra self-supervised task in a multi-task framework can enhance the adversarial robustness of supervised models. In a similar vein, Carmon et al. (2022) discovered that using additional unlabeled data also strengthens the model's adversarial resilience. Furthermore, Chen et al. (2020a) created robust variants of pretext-based SSL tasks, showing that their integration with robust fine-tuning leads to a notable increase in robustness compared to standard adversarial training.

Chhipa et al. (2023) demonstrates a clear relationship between the performance of learned representations within SSL paradigms and the severity of distribution shifts and corruptions and highlights the critical impact of distribution shifts and image corruptions on the performance and resilience of SSL methods. Similarly, Zhong et al. (2022) conduct robustness tests to assess the behavioral differences between contrastive and supervised learning under changes in downstream or pre-training data distributions, while also exploring the effects of data augmentation and feature space characteristics. Kowalczuk et al. (2024) conducts a comprehensive empirical evaluation of the adversarial robustness of self-supervised vision encoders across multiple downstream tasks, revealing the need for broader enhancements in encoder robustness.

## 3 EXPERIMENTAL SETUP

### 3.1 SSL MODELS

While numerous SSL approaches have been proposed Ozbulak et al. (2023), we focus exclusively on the following well-known SSL models because of computational constraints: Barlow Twins Zbontar et al. (2021), BYOL Grill et al. (2020), DINO Caron et al. (2021), MoCoV3 Chen* et al. (2021), SimCLR Chen et al. (2020b), SwAV Caron et al. (2020), and VICReg Bardes et al. (2022). We utilize ResNet50 He et al. (2015) models by default, as most models are trained exclusively in this format. Our experiments utilize the best publicly available ImageNet checkpoints from these models. However, we carried out linear evaluation on Barlow Twins and VICReg since only the backbone weights are available. We used the official repositories for these models for the linear evaluation, but this led to a 2% decrease in performance Furthermore, we assess a supervised baseline for comparison, a standard pre-trained ResNet50 model obtained from the PyTorch library Paszke et al. (2019). All models feature 23.5 million parameters in their backbones and were pre-trained on

Table 1: Performance of various models on ImageNet, Transfer Learning, Segmentation, and Detection tasks, showing both original (Orig.) and adversarial (Adv.) score. The percentage drop in performance from original to adversarial is indicated in red. More detailed results of ImageNet in B.1, transfer learning in B.7, segmentation in B.2, and detection in B.3.

| Model | ImageNet | | Transfer Learning | | Segmentation | | Detection | |
|---|---|---|---|---|---|---|---|---|
| | Orig. | Adv. | Orig. | Adv. | Orig. | Adv. | Orig. | Adv. |
| Barlow Twins | 71.2 | 38.6 ↓46% | 80.3 | 40.1 ↓50% | 76.9 | 20.5 ↓73% | 88.4 | 21.9 ↓75% |
| BYOL | 74.6 | 45.9 ↓39% | 78.7 | 47.3 ↓40% | 76.7 | 19.0 ↓75% | 87.4 | 17.3 ↓80% |
| DINO | 75.3 | 35.1 ↓53% | 80.7 | 35.6 ↓56% | 77.0 | 18.9 ↓76% | 87.6 | 22.0 ↓75% |
| MoCoV3 | 74.6 | 41.3 ↓45% | 80.5 | 42.1 ↓47% | 76.2 | 19.9 ↓74% | 87.3 | 18.5 ↓79% |
| SimCLR | 68.9 | 32.8 ↓52% | 73.1 | 32.6 ↓55% | 75.6 | 19.3 ↓74% | 88.3 | 15.4 ↓82% |
| Supervised | 76.1 | 31.8 ↓58% | 74.6 | 26.6 ↓64% | 74.2 | 16.5 ↓78% | 86.1 | 18.0 ↓79% |
| SwAV | 75.3 | 39.3 ↓48%. | 79.2 | 35.7 ↓55% | 76.5 | 19.2 ↓75% | 86.6 | 20.5 ↓76% |
| VICReg | 71.3 | 38.5 ↓46% | 79.9 | 39.9 ↓50% | 77.9 | 20.5 ↓74% | 88.4 | 14.0 ↓84% |

the ImageNet Russakovsky et al. (2015) training set, containing 1.28 million images, with only the supervised baseline utilizing labels.

## 3.2 IMAGENET AND TRANSFER LEARNING

We use the benchmark suite introduced in the transfer learning study Huh et al. (2016), which encompasses the target datasets like FGVC Aircraft Maji et al. (2013), Caltech-101 Fei-Fei et al. (2004), Stanford Cars Krause et al. (2013), CIFAR 10 Krizhevsky (2009), CIFAR 100 Krizhevsky (2009), DTD Cimpoi et al. (2013), Oxford 102 Flowers Cimpoi et al. (2013), and Food-101 Bossard et al. (2014). We follow Ericsson et al. (2021) for linear evaluation of these datasets. We conducted only linear evaluation because the backbone remains frozen during this process, allowing for a more equitable comparison of objectives within this setup.

For both ImageNet and transfer learning, we apply the same adversarial techniques: Instance Adversarial Attacks (IAA) and Universal Adversarial Perturbations (UAP). In brief, instance-based methods generate unique perturbations for each individual image, while UAP involves creating a single perturbation that applies across the entire dataset. Given the variety of attacks used, further details are provided in Appendix A.1.1, A.1.2, and A.2.

## 3.3 SEGMENTATION

For segmentation, we use only the Pascal VOC 2012 dataset Everingham et al. and train a DeepLabV3+ model Chen et al. (2018a). To conduct the attacks, we follow the setup from Rony et al. (2023), utilizing Alma Rony et al. (2023), Asma Rony et al. (2023), DAG Xie et al. (2017), DDN Rony et al. (2023), FGSM Goodfellow et al. (2014), FMN Pintor et al. (2021), and PGD Madry et al. (2017). While our primary metric is the mean Intersection Over Union (IOU), we also report the Attack Pixel Success Rate (APSR) introduced by Rony et al. (2023). Although our main focus is on using a frozen backbone, we also perform training following the standard procedure.

## 3.4 DETECTION

For object detection, we utilized the INRIA Person Dalal & Triggs (2005) dataset and trained a Faster R-CNN Ren et al. (2016). To perform adversarial attacks, we followed the setup described by Huang et al. (2023), employing the Transfer-based Self-Ensemble Attack (T-SEA). The T-SEA attack can be deployed using various methods and optimizers. In our experiments, we employed BIM Huang et al. (2023), MIM Dong et al. (2018a), PGD Madry et al. (2017), and Optim Huang et al. (2023) methods. Additionally, we explored simpler methods that rely on common optimizers, such as Adam Kingma & Ba (2017), SGD, and Nesterov Nesterov (1983). Throughout our evaluation, we report the mean average precision (mAP) scores as the primary performance metric. While our primary focus was on employing a frozen backbone, we also conducted training experiments following the standard training procedures for comparative analysis.

## 4 RESULTS AND DISCUSSION

In this section, we present our experimental findings on ImageNet, transfer learning, and detection, and discuss each in turn. While we address the results individually, the full detailed results are provided in Appendix B.

### 4.1 IMAGENET

#### 4.1.1 SSL VS SUPERVISED

Most robustness studies on contrastive learning Ho & Vasconcelos (2020); Kim et al. (2020); Jiang et al. (2020); Xie et al. (2020); Zhang et al. (2022); Nguyen et al. (2022) focus on small datasets like CIFAR10 Krizhevsky (2009) and primarily evaluate robustness using adversarial attacks such as FGSM Goodfellow et al. (2014) and PGD Madry et al. (2017). While this is reasonable given that many proposed defenses struggle to scale to larger datasets like ImageNet Russakovsky et al. (2015) due to computational demands, the evaluation process still has a limitation: the infrequent use of UAP. However, since our goal is to assess robustness rather than develop a new defense, this limitation is less relevant for us. To achieve this, we evaluate the robustness of seven different SSL models, as well as a supervised model, against both IAA and UAP.

Our findings, summarized in Tables 3.1 and B.1, show that all SSL models demonstrate higher robustness compared to the supervised model, both in terms of final performance and the drop in adversarial accuracy. This differs from Gupta et al. (2022) which suggests that contrastive learning approaches, like SimCLR and MoCoV3, underperform relative to supervised learning. Their reasoning is that false negative pairs in contrastive SSL lead to instance-level uniformity, weakening class separation in the feature space and making models more susceptible to adversarial attacks. They also argue that SwAV maintains uniformity in its representation space, which similarly contributes to this weakening. However, this doesn't fully apply to MoCoV3, which shows the highest adversarial robustness when paired with ResNet which we further discuss in section 4.1.2. It's important to note that their MoCoV3 assessment is based only on testing the ViT version, which they state it performs worse than both DINO and the supervised model that are both ViT. Additionally, they claim that non-contrastive methods like DINO and BYOL are not impacted by the same limitations as contrastive learning. Yet, in our case, DINO with ResNet shows the weakest adversarial robustness score on IAA, though their evaluation focuses on the ViT variant. We provide a more detailed discussion of this in section 4.1.4.

Furthermore, the presence of UAP exposes significant weaknesses in the supervised model, as shown in Figure 2, illustrating how it alters the robustness compared to IAA and influences the overall average. In contrast, SSL models like SimCLR and DINO, despite facing challenges, perform notably better. Notably, SwAV, which ranks as the second-worst model in IAA, emerges as the second-best overall and BYOL significantly outperforms other models on UAP and maintains its lead even when combined with IAA. Overall, our findings emphasize that the diversity and type of attacks are critical when evaluating the adversarial robustness of SSL models and comparing them against supervised model. Moreover, the distinction between contrastive and non-contrastive approaches doesn't fully hold, as there is at least one model from each category that challenges the conclusion from Gupta et al. (2022) that non-contrastive methods are more robust due to their exclusion of negative samples in the loss function.

#### 4.1.2 WHAT MAKES MOCOV3 ROBUST?

Although MoCoV3 and SimCLR both utilize the InfoNCE Sohn (2016); van den Oord et al. (2019) objective, there is a notable difference in their adversarial robustness and baseline accuracy. To understand this disparity, we assess the adversarial robustness of MoCoV1 He et al. (2019) and MoCoV2 Chen et al. (2020c), aiming to identify the enhancements responsible for this effect. Full results of MoCo experiments are in Appendix B.6.

**A brief MoCo History**. *MoCoV1 introduced the idea of using a dynamic dictionary with a queue and a momentum-updated encoder to improve the quality of learned representations. This approach addresses the challenge of negative sample mining in contrastive learning by maintaining a large and consistent set of negative samples over time. MoCoV2 builds on this by incorporating simple*

*architectural improvements, such as using a multi-layer projection head and stronger data augmentation techniques. MoCoV3 enhances MoCoV1 and V2 by removing the memory bank, as large batch sizes reduce the need for it. Additionally, it incorporates a prediction head similar to those in BYOL and SimSiam Chen & He (2020).*

MoCoV2 achieves its most significant improvement over MoCoV1 primarily due to the introduction of a non-linear projector, resulting in a 10% performance increase, while stronger augmentation yields only a marginal benefit. We observe that MoCoV2 shows slight improvements over MoCoV1 in terms of IAA attacks, but it demonstrates significant advancements against UAP attacks. It could be argued that this highlights the subpar representations learned in MoCoV1, rather than being solely due to the projection head's output. Ibrahim et al. (2024) suggest that a non-linear projector isn't always essential for acquiring effective representations. However, given that a strong model without projections has yet to be established, it appears that projections are crucial for enhancing both performance and adversarial robustness.

The enhancement in MoCoV3's performance over MoCoV2 primarily stems from the introduction of the prediction head in the query encoder and the use of a larger batch size. Unlike MoCoV2, MoCoV3 shows significant improvements in both IAA and UAP, highlighting the prediction head's critical role in the robustness of MoCoV3. Momentum appears to be a common feature in robust models such as MoCoV3 and BYOL, whereas MoCoV2 exhibits performance similar to SimCLR.

### 4.1.3 AUGMENTATIONS VS ALGORITHMS

Morningstar et al. (2024) demonstrate that, in their analysis of several popular SSL methods, many algorithmic improvements, such as prediction networks or new loss functions, had minimal impact on downstream task performance. In contrast, stronger augmentation techniques resulted in more significant performance gains. Their findings challenge the view that SSL progress is primarily driven by algorithmic advancements and suggest that augmentation diversity, along with data and model scale, are more critical to recent advancements in SSL.

This complicates the comparison because we lack controlled baselines for the augmentations across different objectives. For instance, when examining the robustness of MoCoV3 relative to V2, it suggests the importance of the prediction head, but it's important to acknowledge a slight variation in augmentation, the impact of which is unclear. Despite this, the noticeable drop in accuracy across objectives indicates that algorithmic innovations do play a role in adversarial robustness, as a higher performance score doesn't always equate to improved robustness on ImageNet.

### 4.1.4 RESNET VS VIT IN ADVERSARIAL ROBUSTNESS

While ViTs are generally seen as more robust than CNNs Naseer et al. (2020), Pinto et al. (2022); Bai et al. (2021) demonstrate that with the right training methods, CNNs Lecun et al. (1998) can achieve comparable robustness. Despite ViT's success Dehghani et al. (2023); Dosovitskiy et al. (2021); Chen* et al. (2021); Caron et al. (2021); Oquab et al. (2024), most SSL methods still use ResNet for validation. For this reason, we focus on MoCoV3 and DINO, as they are the only models that include ViT training. Additionally, we focus exclusively on the smaller versions of these models, which have parameter counts comparable to ResNet50 and we share all results of ViT vs ResNet in Appendix B.4. As previously noted in Section 4.1.1, there is a notable difference in adversarial performance between ResNet and ViT. Specifically, MoCoV3 performs worse with ViT, while DINO achieves strong results, though it shows weaker performance with ResNet.

There are two key algorithmic differences between MoCoV3 and DINO: the presence of a prediction network and the structure of the SSL objective. MoCoV3 includes a prediction network, while DINO does not, even though other distillation-based methods rely on it to avoid collapse. MoCoV3 uses the standard InfoNCE objective, whereas DINO employs a distinct approach. DINO centers the student network's output using a running mean to minimize sensitivity to mini-batch size and applies a softmax to discretize the representations smoothly. Balestriero et al. (2023) argue that the softmax-based discretization in DINO functions as an online clustering mechanism, where the final layer before the softmax contains clustering prototypes and their corresponding weights. As a result, the output of the penultimate layer is clustered using the weights of the final layer. Furthermore, DINO uses multi-crop augmentation similar to SwAV. With this, DINO becomes very similar to SwAV which uses Sinkhorn-Knopp Cuturi (2013) clustering instead.

We note that both SwAV and DINO demonstrate brittleness on IAA, with SwAV showing a marked improvement over DINO on UAP. This suggests that clustering methods, whether implicit (DINO) or explicit (SwAV), are fragile when applied to IAA, while DINO faces significant challenges with UAP. Conversely, DINO-ViT emerges as the most robust model for IAA and also performs better on UAP than ResNet. However, MoCo's findings are contrary to those observed with DINO, complicating the assessment of architectural robustness. It's important to highlight that MoCo-ViT was only trained for 300 epochs, whereas DINO was trained for 800 epochs. This discrepancy is notable, as ViT is inherently computationally demanding, which may lead to brittleness due to undertraining. Unfortunately, without multiple checkpoints for these models at various epochs, we are unable to evaluate this further.

### 4.1.5 IMPACT OF TRAINING DURATION

SSL models tend to demonstrate better performance as training epochs increase Chen et al. (2020b); Chen* et al. (2021); Caron et al. (2020). However, due to computational constraints, many models are reported with different numbers of epochs. This prompts the question of whether longer training durations enhance or reduce adversarial robustness. As noted earlier in section 4.1.4, ViT models do not have checkpoints at various epochs, so we instead focus on ResNet-based SSL models, specifically SwAV and MoCoV3, which offer multiple checkpoints throughout the training process and full results are in Appendix B.5

We find that both SwAV and MoCo show a modest improvement of 1% on IAA across various epochs, which is minimal compared to the rise in original accuracy. In contrast, both methods exhibit a significant increase in UAP after surpassing 100 epochs, with the 200 and 300-epoch checkpoints in SwAV and MoCo aligning well with the best-performing models. Overall, our results suggest that despite differences in reported checkpoints, robustness generally remains stable or slightly improves during training, reinforcing our earlier analysis, even when models are trained for varying numbers of epochs.

### 4.2 TRANSFER LEARNING

A key question is whether robustness on ImageNet correlates with robustness on other classification datasets. We present the averaged total results in Table 3.1, along with combined scores that differentiate by attack type, as well as individual dataset results in AppendixB.7. Our results show a strong correlation, with a coefficient of 0.97. Notably, most models achieve similar transfer learning performance, except for Supervised and SimCLR, supporting the conclusions of Ericsson et al. Despite a significant performance gap between SimCLR and Supervised on ImageNet, Supervised not only ranks second-lowest but is also the least robust overall, indicating that SSL models better transfer their robustness from ImageNet to other datasets.

On IAA, VICReg, Barlow Twins, BYOL, and MoCoV3 exhibit similar levels of robustness, while DINO, SimCLR, SwAV, and supervised lag behind, though the performance gap is narrower compared to ImageNet. The most striking differences emerge under UAP, where BYOL significantly outperforms others, and Supervised performs poorly, with a 17% deficit compared to DINO and SimCLR, the next least robust models. Overall, our findings confirm that robustness on ImageNet translates well to other datasets.

### 4.3 SEGMENTATION AND DETECTION

Both the ImageNet and transfer learning experiments have so far focused on linear evaluation across various datasets with a frozen backbone, which helps to capture differences between different SSL models. However, tasks like segmentation and detection are inherently different from object recognition, not just in nature but also in their experimental setups. These tasks require adding multiple modules to adapt ResNet or other vision backbones, which leads to a substantial increase in the number of parameters, often nearly doubling the size of ResNet. Therefore, studying how different SSL models perform in these alternative setups, beyond typical classification, becomes particularly intriguing. Segmentation and Detection results are in table 3.1 with ImageNet and Transfer Learning and their individual scores are in Appendix B.2 and B.3 respectively.

**Segmentation**

Unlike in classification, we didn't observe a strong correlation between ImageNet robustness and segmentation performance which. One notable point is that the supervised model performs slightly worse than others, including in terms of robustness, though the differences are small, making it difficult to draw definitive conclusions. A similar argument applies to the APSR scores. One possible explanation for this is that adversarial attacks may target the segmentation modules more than the backbones, which make up a large portion of the overall model and could be enough to cause incorrect predictions.

Since freezing the backbone isn't the standard practice for training segmentation models, we also tested SSL models with the backbone unfrozen. Interestingly, the clean scores were generally lower than with a frozen backbone, except for the Supervised model. This is because our reproduction of the Supervised model performed significantly worse than the available checkpoints, so we used the standard segmentation model from MMSegmentation Contributors (2020). Despite this, our findings were similar to the frozen backbone case, though SimCLR performed slightly worse. Overall, these experiments suggest that the adversarial robustness of segmentation models has almost no reliance on the backbone, meaning SSL models have virtually no effect on the final robustness. This contrasts with object recognition, where we observe significant differences between different SSL objectives.

**Detection**

The observations for detection closely mirror those for segmentation, highlighting that robustness in ImageNet does not necessarily indicate robustness in detection tasks. However, there are some important distinctions from the segmentation analysis. With the frozen backbone, we find VICReg to be the least robust, which strongly contradicts our earlier findings in recognition and segmentation. In contrast, Barlow Twins continues to perform well and maintains a reasonable level of robustness across various objectives. DINO and SwAV also show respectable performance, even though we previously identified them as fragile on ImageNet. In standard model training with an unfrozen backbone, the supervised model exhibits significantly lower robustness. In summary, the intricate models designed for various tasks significantly influence performance, reducing the importance of the backbone and making it more challenging to extend our analysis to these downstream tasks.

## 5 CONCLUSIONS

In essence, our exploration of the adversarial robustness of SSL models suggests that these models generally outperform their supervised counterparts, particularly in ImageNet classification and transfer learning tasks. However, we recognize that their robustness is less pronounced in segmentation and detection tasks. Our findings indicate that architectural choices can influence robustness, though the extent of this impact varies depending on the SSL objective used. Additionally, while extending training durations may provide slight improvements in robustness, the benefits appear limited. Overall, this study highlights the need for further research into enhancing the adversarial robustness of visual SSL systems. We hope our findings contribute to the ongoing dialogue in this area and encourage future investigations aimed at developing more resilient models in complex environments.

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

# A APPENDIX

## A.1 ADVERSARIAL ATTACKS

### A.1.1 INSTANCE ADVERSARIAL ATTACKS

Instance adversarial methods, or per-instance generation, involve crafting distinct perturbations for each individual image within the dataset on which the model has been trained or fine-tuned. The generation of these perturbations relies on various techniques, which are determined by the specific goals of the attack, the level of access granted to the model—such as full access to model weights, predictions alone, or prediction scores (logits)—and the distance metrics employed. While multiple classification schemes for adversarial attacks exist, we adopt the widely accepted taxonomy for clarity and consistency.

White-box attacks, in this context, presume complete access to the model, including its architecture and parameters. The primary approach utilizes the gradients derived from the loss function to generate adversarial perturbations. These perturbations are then applied to the image within the constraints of specific distance metrics, such as $l_0$, $l_1$, $l_2$, or $l_\infty$. Specifically, $l_0$ measures the number of altered pixels, $l_1$ quantifies the absolute difference between images, $l_2$ computes the Euclidean distance, and $l_\infty$ captures the magnitude of the largest perturbation applied to any pixel.

Gradient-based methods exploit the gradient of the neural network's loss function with respect to the input data, strategically altering the input to increase the loss and induce misclassification. The foundational work in this domain is attributed to the Fast Gradient Sign Method (FGSM) Goodfellow et al. (2014), which represents the first successful application of gradient-based adversarial perturbations. Over time, iterative approaches such as I-FGSM/BIM Kurakin et al. (2018) and momentum-based techniques like MI-FGSM Dong et al. (2018b) have been introduced to enhance the effectiveness of these perturbations, particularly for classification tasks. However, these methods often exhibit limited transferability to other models, a key challenge in black-box settings Madry et al. (2017); Dong et al. (2019).

Some studies suggest that sharp curvatures around data points can obscure the true direction of steepest ascent, reducing the success of cross-model transferability in adversarial attacks. To address this issue, methods such as the R-FGSM algorithm introduce random perturbations to the single-step FGSM algorithm, allowing a small step in the loss space to discover more generalizable and robust perturbations that may effectively transfer to other models Tramèr et al. (2017).

Building on techniques designed to improve model generalization, several methods have been developed specifically to enhance cross-model transferability. For instance, Lin et al. (2019) introduces NI-FGSM and SINI-FGSM, which leverage Nesterov momentum to avoid suboptimal local maxima. The look-ahead property of Nesterov momentum, combined with the "scale-invariant" property of deep neural networks (as detailed in their paper), helps mimic the effect of an ensemble model by using loss-preserving data augmentation. Similarly, Wang & He (2021) establishes a connection between model generalization and the cross-model transferability of adversarial examples, proposing VMI-FGSM, a more stable update algorithm. VMI-FGSM calculates the variance of the gradient by sampling multiple examples from the neighborhood of a data point, refining the gradient to produce more stable perturbations. This method can be extended to more complex attacks, as demonstrated with VNI-FGSM in the same work Wang & He (2021). Likewise, PI-FGSM and PI-FGSM++ modify the gradient update rule by focusing on patch-based rather than pixel-wise perturbations Gao et al. (2020a;b). DI-FGSM, as discussed in relation to SINI-FGSM Lin et al. (2019), employs random padding and resizing operations to enhance data input for auxiliary models Xie et al. (2019). TAP also tries to increase cross-model transferability by introducing distance maximization between intermediate feature maps of the adversarial and benign datapoints. It also regularize the images to reduce high frequency perturbations as they claim Convolution may act as a smoother, and it will increase the black-box transferability performance of perturbation Zhou et al. (2018).

Improving the transferability of per-instance attacks can, however, lead to reduced effectiveness against auxiliary models, and vice-versa Tramèr et al. (2017); Gao et al. (2020a). Therefore, various strategies have been proposed to optimize attack performance based on the level of access to the target model.

In contrast, optimization-based attacks approach the generation of adversarial examples as an optimization problem, where a specific objective is minimized subject to given constraints. While gradient-based methods update images directly using gradient information and typically rely on the $l_\infty$ norm as a boundary, optimization-based methods employ a more formal problem definition that allows for the use of advanced optimization techniques such as L-BFGS. Consequently, the $l_2$ norm is frequently utilized in these methods alongside other $l$ norms.

The first demonstration of adversarial examples by Szegedy et al. (2013) employed the L-BFGS method to identify images within an $l_2$ ball that were visually similar to the original image. Similarly, Carlini & Wagner (2017) modified the original minimization problem—focusing on minimizing the distance between adversarial examples and the original data points across several $l$ norms—to develop the CW attack, one of the most prominent adversarial attack methods, which also leverages L-BFGS for optimization.

On the other hand, Projected Gradient Descent (PGD) employs an iterative approach, projecting updates back onto the $l_\infty$ ball of the original data point to generate adversarial perturbations Madry et al. (2017). The key distinction between PGD and other iterative gradient-based methods, such as FGSM variants, lies in the fact that PGD treats each iteration as a solution to the same optimization problem. PGD ensures that each iterative step remains within the neighborhood of the original data point, while iterative FGSM methods use the newly generated steps to continue further processing.

The EADL1 and EADEN attacks adopt a similar approach to the CW attack but introduce a modification to the loss function by incorporating an additional $l_1$ distance term in the minimization problem. The $l_1$ distance, which measures the total variation of the perturbation, promotes sparsity in the adversarial perturbation. While sparsity is not widely employed in adversarial example generation, it is commonly used in image denoising and restoration techniques. These methods utilize the Iterative Shrinkage-Thresholding Algorithm (ISTA) to solve the corresponding optimization problem Chen et al. (2018b).

As with gradient-based methods like FGSM, several improvements have been made to optimization-based methods to address specific needs, with a particular focus on enhancing PGD Madry et al. (2017). For example, PGD-$l_2$ incorporates the $l_2$ norm instead of the $l_\infty$ norm to better fool target models Madry et al. (2017), while TPGD replaces the Cross-Entropy loss in PGD with KL-Divergence to optimize the perturbation process Zhang et al. (2019). Additionally, Auto-PGD modifies the step size in PGD within a budget-aware context, arguing that the original PGD method does not account for trends that lead to more effective adversarial perturbations Croce & Hein (2020).

The Jitter attack introduces a novel objective function for adversarial perturbation generation, departing from the conventional Cross-Entropy objective. The study suggests that many adversarial attacks predominantly fool a limited set of classes rather than broadly deceiving the entire model. The proposed objective seeks to enhance the fooling rate across a wider range of classes, aiming for more generalized misclassification Schwinn et al. (2023).

Additionally, there are gradient-free approaches that remain relatively underexplored. For instance, the Simultaneous Perturbation Stochastic Approximation (SPSA) method estimates gradients by perturbing the input in random directions, enabling the approximation of gradients for objectives that cannot be differentiated analytically. This approach offers deeper insights into the model's behavior, with the paper also claiming that the stochastic perturbations introduced by sampling allow algorithms to converge toward a global minimum Uesato et al. (2018).

While white-box attacks exploit full access to the model, this is often not a realistic scenario. In many cases, model weights are not shared, or gradient information is unavailable. Although efforts have been made to enhance cross-model transferability, as discussed previously, there are also specific attack schemes designed to target models in black-box settings. For example, the Square Attack leverages random search combined with model scores—probability distributions over class predictions—to generate perturbations. In essence, the algorithm makes random modifications to the input data and retains changes that yield progress toward the objective function Andriushchenko et al. (2020).

Among black-box attacks, some methods focus on $l_0$ norm-based perturbations. Pixle, for instance, is a black-box attack that utilizes random search and the $l_0$ norm, altering a small number of pixels to generate adversarial examples Pomponi et al. (2022). On a more constrained scale, the OnePixel

attack modifies only a single pixel, maintaining an $l_0$ norm of 0, and despite its simplicity, it is capable of fooling models to some extent. However, it is less effective than other methods due to its significant restrictions. This raises important questions about our understanding of Deep Neural Networks and their vulnerability to minimal perturbations Su et al. (2019).

### A.1.2 Universal Adversarial Perturbations

The Universal Adversary (UAP) represents a singular perturbation crafted for an entire image dataset. The rationale behind UAP is to identify a perturbation, subject to specified constraints, capable of deceiving the model across a majority of images in the dataset, as initially demonstrated by Moosavi-Dezfooli et al. (2017), which utilizes DeepFool to create an average perturbation for the entire dataset. It has been empirically observed that universal adversaries exhibit heightened transferability across diverse models and datasets compared to instance methods. UAP's are important as they are independent from the input - to some extend - they reveal intrinsic chracterics of models of interest Chaubey et al. (2020); Ye et al. (2023).

Two primary techniques are employed for crafting UAPs: (1) generation with generative models, as evidenced by works such as Hayes & Danezis (2018); Mopuri et al. (2018b), and (2) learning a perturbation designed to disrupt the representations acquired by the models.

UAPs can be further categorized into two classes: data-dependent attacks, which require a comprehensive and general dataset that the attacker seeks to compromise (e.g., ImageNet), and data-independent attacks, which do not rely on any specific dataset.

The first example of UAP, referred to here as UAP-DeepFool (to avoid confusion with the broader class of UAP attacks), utilizes the DeepFool per-instance adversarial attack method which computes perturbations by manipulating the geometry of decision boundaries. UAP-DeepFool iteratively determines the worst-case direction for each data point, and aggregating the results into a universal perturbation - if it is succesfull -, which is then projected onto an $l_\infty$ ball Moosavi-Dezfooli et al. (2017). Following this work, UAPEPGD replaces the DeepFool approach with Projected Gradient Descent (PGD), an optimization-based adversarial attack method, to craft stronger adversarial examples Deng & Karam (2020).

ASV - to our best knowledge - is the first UAP that does not require label information, relying solely on images to generate UAPs. Adversarial Semantic Vectors (ASVs) represent one of the first UAP methods that do not require label information, relying solely on images to generate UAPs. The study suggests that since adversarial perturbations typically exhibit small magnitudes, perturbations in the non-linear maps computed by deep neural networks (DNNs) can be approximated using the Jacobian matrix Khrulkov & Oseledets (2018). Similarly, the STD (Dispersion Reduction) attack seeks to reduce the "contrast" of the internal feature map by targeting the lower layers of Convolutional Neural Networks (CNNs). These lower layers typically detect simple image features such as edges and textures, which are common across datasets and CNN models. By reducing the contrast (measured as the standard deviation of feature maps), the resulting images become indistinguishable to the model Lu et al. (2020).

Self-Supervised Perturbation (SSP) takes a different approach, arguing that adversarial examples generated through gradients using labels fail to capture intrinsic properties of models. SSP aims to maximize "feature distortion," the changes in the network's internal representation caused by adversarial examples compared to the original image, in order to fool subsequent layers in the model Naseer et al. (2020).

FG-UAP builds upon this by exploiting a phenomenon referred to as "Neural Collapse," where, as noted, different class activations converge to class means, allowing a single common perturbation to fool the model across a wide range of images. This collapse happens primarily in the final layers of the model, and FG-UAP targets these regions to generate effective UAPs Ye et al. (2023).

Another label-independent UAP method, L4A, focuses on the success of adversarial perturbations during cross-finetuning. L4A targets the lower layers of models, which remain more stable during finetuning (as they detect simple features), and utilizes the Frobenius norm for optimization, with variants such as L4A-base, L4A-fuse, and L4A-ugs. L4A-base attacks the lowest layer, L4A-fuse attacks lowest 2 layers and L4A-ugs uses samples from a Gaussian distribution where mean and standard deviation is in close range of downstream task Ban & Dong (2022).

Data-independent UAP methods do not utilize any dataset for adversarial perturbation generation, instead focusing on the intrinsic characteristics of models. Fast Feature Fool (FFF) was the first adversarial attack method that did not use a dataset. It aims to disrupt the features learned at individual CNN layers, proposing that non-discriminative activations can lead to eventual misclassification. FFF over-saturates the learned features at multiple layers, misleading subsequent layers in the network Mopuri et al. (2017). Following that work GD-UAP, changes the objective a little bit and add other variations such as "mean-std" and "sampled" versions to improve perturbation performance. The "mean-std" variant uses the mean and standard deviation of the test dataset to better align perturbations with dataset characteristics to prevent perturbation dataset mismatch, while the "sampled" version employs a small sample from the dataset to capture its statistics and semantics Mopuri et al. (2018a). In our work, we have also integrated "mean-std" and "one-sample" versions of GD-UAP to FFF, since they are highlt similar as GD-UAP is a follow-up work FFF. PD-UAP, another data-independent method, focuses on predictive uncertainty rather than any specific image data, aligning perturbations with task-specific objectives Mopuri et al. (2017).

To accommodate both Vision Transformers (ViTs) and ResNets, we have adapted some of these attacks, originally designed for CNNs, to work with ViTs. For low-level layer attacks, we applied them to the first few blocks of the ViT model, following methods like SSP and L4A. For FFF, which typically uses mean of ReLU activations and a logarithmic operation, we modified the procedure to suit ViTs, which employ GeLU activations (capable of taking values below zero), by applying an absolute value operator between the mean and logarithmic functions. In conducting these experiments, we strove to maintain fair comparisons and minimized the introduction of tweaks to the original methodologies.

## A.2 FGSM AND PGD VERSIONS

| Attack Version | Attack Type | $\varepsilon$ | Step Count | Norm |
|:---:|:---:|:---:|:---:|:---:|
| $FGSM_1$ | FGSM | 0.25 | - | $\infty$ |
| $FGSM_2$ | FGSM | 1 | - | $\infty$ |
| $PGD_1$ | PGD | 0.25 | 20 | $\infty$ |
| $PGD_2$ | PGD | 1 | 20 | $\infty$ |
| $PGD_3$ | PGD | 0.25 | 40 | $\infty$ |
| $PGD_4$ | PGD | 1 | 40 | $\infty$ |
| $PGD_5$ | PGD | 0.5 | 40 | $\|\cdot\|_2$ |

Table 2: Hyperparameters of the different FGSM and PGD attacks that we use in ImageNet and transfer learning.

# B  FULL RESULTS

## B.1  IMAGENET

Table 3: This table presents the results of various instance and universal adversarial perturbation (UAP) attacks on the Imagenet-1k dataset, with all UAP attack names in *italics*. Different configurations of FGSM and PGD are denoted, such as $FGSM_1$ and $PGD_1$. Average results for universal adversarial perturbations (UAP Avg.), instance adversarial attacks (IAA Avg.), and overall adversarial performance (Adv Avg.) are reported at the bottom, including percentage drops relative to clean accuracy.

| | Barlow | BYOL | DINO | MoCoV3 | SimCLR | Supervised | SwAV | VICReg |
|---|---|---|---|---|---|---|---|---|
| $FGSM_1$ | 42.41 | 39.41 | 24.68 | 42.67 | 24.29 | 38.83 | 24.71 | 42.42 |
| $FGSM_2$ | 18.11 | 13.47 | 5.66 | 15.53 | 8.84 | 12.18 | 6.35 | 18.11 |
| $PGD_1$ | 42.38 | 39.63 | 25.65 | 42.39 | 26.6 | 35.26 | 26.48 | 42.41 |
| $PGD_2$ | 1.48 | 0.65 | 0.18 | 1.06 | 0.25 | 0.37 | 0.18 | 1.5 |
| $PGD_3$ | 42.6 | 39.82 | 25.85 | 42.56 | 26.79 | 35.39 | 26.73 | 42.6 |
| $PGD_4$ | 1.19 | 0.5 | 0.14 | 0.82 | 0.2 | 0.28 | 0.14 | 1.2 |
| $PGD_5$ | 5.18 | 3.44 | 0.67 | 4.79 | 0.9 | 1.9 | 0.69 | 5.15 |
| DIFGSM | 52 | 52.71 | 41.12 | 54.09 | 42.57 | 51.43 | 45.65 | 52.49 |
| CW | 0.18 | 0.02 | 0 | 0.02 | 0.02 | 0.02 | 0 | 0.19 |
| Jitter | 59.83 | 61.92 | 60.26 | 62.47 | 56.4 | 62.75 | 61.16 | 59.84 |
| TIFGSM | 61.04 | 62.27 | 56.98 | 61.47 | 55.63 | 62.16 | 60.07 | 59.91 |
| PIFGSM | 34.38 | 29.83 | 14.54 | 34.1 | 13.34 | 28.64 | 14.12 | 34.43 |
| EADEN | 0 | 0 | 0 | 0 | 0 | 0 | 0 | 0 |
| OnePixel | 69.34 | 72.5 | 72.83 | 72.64 | 66.47 | 73.27 | 72.73 | 69.38 |
| Pixle | 25.22 | 28.67 | 19.41 | 31.45 | 21.75 | 23.21 | 16.95 | 25.23 |
| SPSA | 66.59 | 69.59 | 68.11 | 69.93 | 63.01 | 69.48 | 68.61 | 66.63 |
| Square | 4.44 | 2.62 | 1.3 | 3.15 | 4.22 | 0.87 | 1.99 | 4.49 |
| TAP | 70.31 | 74.36 | 73.78 | 73.72 | 68.1 | 68.98 | 75.05 | 70.33 |
| *ASV* | 44.9 | 60.98 | 45.08 | 50.21 | 62.67 | 32.83 | 53.66 | 44.86 |
| *FFF (no-data)* | 45.14 | 60.45 | 43.58 | 49.63 | 43.72 | 31.54 | 51.88 | 45.02 |
| *FFF (mean-std)* | 44.64 | 60.7 | 43.58 | 49.01 | 48.75 | 32.69 | 53.4 | 44.69 |
| *FFF (one-sample)* | 45 | 60.88 | 44.5 | 49.9 | 34.38 | 32.15 | 53.33 | 44.97 |
| *FG-UAP* | 42.26 | 56.13 | 37.41 | 45.28 | 3.2 | 27.53 | 44.59 | 42.2 |
| *GD-UAP (no-data)* | 45.04 | 60.66 | 43.71 | 49.41 | 32.91 | 32.05 | 52.19 | 45.01 |
| *GD-UAP (mean-std)* | 44.69 | 60.6 | 43.78 | 49.33 | 55.8 | 32.72 | 53.11 | 44.8 |
| *GD-UAP (one-sample)* | 45.1 | 60.93 | 44.59 | 49.98 | 40.16 | 32.32 | 53.4 | 45.12 |
| *L4A-base* | 44.15 | 60.63 | 44.61 | 49.51 | 9.87 | 32.99 | 49.89 | 44.11 |
| *L4A-fuse* | 44.21 | 60.42 | 44.64 | 49.48 | 9.22 | 32.99 | 49.69 | 44.07 |
| *L4A-ugs* | 44.97 | 61.01 | 45.25 | 49.83 | 56.46 | 32.51 | 53.37 | 44.89 |
| *PD-UAP* | 45.13 | 61.18 | 44.14 | 50.05 | 61 | 32.66 | 53.45 | 45.1 |
| *SSP* | 43.15 | 59.734 | 43.09 | 47.61 | 37.42 | 29.71 | 51.21 | 43.07 |
| *STD* | 44.43 | 60.78 | 44.16 | 49.4 | 51.57 | 32.49 | 53.18 | 44.4 |
| *UAP (DeepFool)* | 45.43 | 61.14 | 45.43 | 50.43 | 24.35 | 33.48 | 53.86 | 45.44 |
| *UAPEPGD* | 45.79 | 61.37 | 45.54 | 50.67 | 64.28 | 33.87 | 54.26 | 45.61 |
| Clean Accuracy | 71.2 | 74.57 | 75.28 | 74.57 | 68.90 | 76.13 | 75.27 | 71.26 |
| IAA Avg. | 33.14 ↓54% | 32.86↓56% | 27.28↓64% | 34.04↓54% | 26.63↓61% | 31.39↓59% | 27.87↓63% | 33.12↓54% |
| UAP Avg. | 44.62 ↓37% | 60.47↓19% | 43.94↓42% | 49.35↓34% | 39.73↓42% | 32.15↓58% | 52.15↓31% | 44.59↓37% |
| Adv Avg. | 38.55 ↓46% | 45.85↓39% | 35.13↓53% | 41.26↓45% | 32.80↓52% | 31.75↓58% | 39.29↓48% | 38.52↓46% |

## B.2 SEGMENTATION

| Metric | Barlow | BYOL | DINO | MocoV3 | SimCLR | Supervised | SwAV | VICReg |
|---|---|---|---|---|---|---|---|---|
| **Alma** | | | | | | | | |
| IOU (↑) | 0.35 | 0.33 | 0.34 | 0.4 | 0.31 | 0.26 | 0.38 | 0.39 |
| APSR (↓) | 99.02 | 99.01 | 99.02 | 98.91 | 99 | 99.01 | 99.01 | 98.99 |
| **Asma** | | | | | | | | |
| IOU (↑) | 49.4 | 63.39 | 61.36 | 61.57 | 32.06 | 77.3 | 62.12 | 50.38 |
| APSR (↓) | 15.39 | 10.95 | 11.38 | 12.18 | 22.78 | 5.29 | 11.56 | 14.48 |
| **DAG** | | | | | | | | |
| IOU (↑) | 0.02 | 0.02 | 0.02 | 0.02 | 0.03 | 0.05 | 0.02 | 0.02 |
| APSR (↓) | 99.87 | 99.91 | 99.89 | 99.88 | 99.83 | 99.74 | 99.89 | 99.89 |
| **DDN** | | | | | | | | |
| IOU (↑) | 5.62 | 4.64 | 5.11 | 7.16 | 1.67 | 1.52 | 6.91 | 4.94 |
| APSR (↓) | 89.66 | 92.6 | 92.75 | 88.01 | 97.24 | 88.56 | 90.77 | 87.23 |
| **FGSM** | | | | | | | | |
| IOU (↑) | 30.35 | 29.28 | 30.41 | 29.43 | 32.15 | 38.31 | 29.4 | 29.84 |
| APSR (↓) | 35.91 | 45.62 | 39.66 | 41.71 | 33.55 | 21.36 | 42.94 | 39.31 |
| **FMN** | | | | | | | | |
| IOU (↑) | 5.4 | 5.29 | 4.86 | 5.19 | 5.07 | 2.74 | 4.9 | 6.2 |
| APSR (↓) | 91.18 | 92.25 | 91.02 | 91.42 | 89.88 | 93.53 | 91.94 | 89.99 |
| **PGD** | | | | | | | | |
| IOU (↑) | 12.67 | 13.16 | 12.75 | 13.06 | 12.88 | 10.92 | 12.98 | 13.04 |
| APSR (↓) | 70.07 | 82 | 77 | 79.27 | 71.15 | 67.4 | 77.31 | 72.43 |
| **Clean IOU (↑)** | 72.63 | 70.37 | 71.65 | 71.25 | 71.96 | 77.35 | 70.8 | 70.33 |
| **Clean APSR (↓)** | 7.18 | 8.29 | 7.64 | 7.83 | 7.2 | 5.27 | 8.21 | 8.01 |
| **Adversarial IOU (↑)** | 14.83↓80% | 16.59↓78% | 16.41↓77% | 16.69↓77% | 12.02↓83% | 18.73↓76% | 16.67↓77% | 14.97↓79% |
| **Adversarial APSR (↓)** | 71.59↑64% | 74.62↑66% | 72.96↑65% | 73.05↑65% | 73.35↑66% | 67.84↑64% | 73.35↑65% | 71.76↑64% |

Table 4: Performance metrics (IOU and APSR) for various self-supervised and supervised models under different adversarial attacks, using unfrozen backbones. Clean and adversarial scores are reported, with percentage changes in adversarial performance noted. Higher IOU and lower APSR indicate better results

| Metric | Barlow | BYOL | DINO | MocoV3 | SimCLR | Supervised | SwAV | VICReg |
|---|---|---|---|---|---|---|---|---|
| **Alma** | | | | | | | | |
| IOU (↑) | 0.39 | 0.31 | 0.37 | 0.37 | 0.55 | 0.28 | 0.35 | 0.41 |
| APSR (↓) | 99.02 | 99.02 | 99.02 | 99.02 | 98.45 | 99.01 | 99.02 | 99.02 |
| **Asma** | | | | | | | | |
| IOU (↑) | 76.06 | 72.84 | 75.32 | 72.84 | 70.42 | 69.84 | 74.09 | 76.74 |
| APSR (↓) | 6.01 | 7.23 | 6.14 | 7.58 | 5.98 | 8.18 | 6.75 | 5.98 |
| **DAG** | | | | | | | | |
| IOU (↑) | 0.03 | 0.04 | 0.02 | 0.04 | 0.04 | 0.02 | 0.03 | 0.03 |
| APSR (↓) | 99.90 | 99.87 | 99.89 | 99.87 | 99.82 | 99.87 | 99.88 | 99.89 |
| **DDN** | | | | | | | | |
| IOU (↑) | 10.81 | 9.76 | 6.91 | 10.74 | 6.62 | 2.95 | 8.57 | 11.12 |
| APSR (↓) | 79.62 | 75.93 | 82.58 | 78.71 | 75.20 | 87.30 | 83.48 | 80.41 |
| **FGSM** | | | | | | | | |
| IOU (↑) | 35.16 | 31.90 | 30.88 | 35.18 | 36.25 | 27.70 | 32.37 | 34.99 |
| APSR (↓) | 33.29 | 33.63 | 36.12 | 33.63 | 27.35 | 36.99 | 36.10 | 33.69 |
| **FMN** | | | | | | | | |
| IOU (↑) | 6.63 | 6.23 | 6.22 | 6.42 | 8.92 | 4.23 | 6.48 | 6.56 |
| APSR (↓) | 87.73 | 87.10 | 87.12 | 87.70 | 81.28 | 91.30 | 87.23 | 87.23 |
| **PGD** | | | | | | | | |
| IOU (↑) | 14.13 | 12.12 | 12.12 | 13.25 | 12.23 | 10.49 | 12.31 | 13.51 |
| APSR (↓) | 76.16 | 75.49 | 75.49 | 76.60 | 73.38 | 78.37 | 80.82 | 77.62 |
| **Clean IOU (↑)** | 76.90 | 76.69 | 77.01 | 76.19 | 75.62 | 74.20 | 76.54 | 77.89 |
| **Clean APSR (↓)** | 5.75 | 5.74 | 5.38 | 6.01 | 5.98 | 6.35 | 5.79 | 5.48 |
| **Adversarial IOU (↑)** | 20.46↓73% | 19.03↓75% | 18.83↓76% | 19.83↓74% | 19.29↓74% | 16.50↓78% | 19.17↓75% | 20.48↓74% |
| **Adversarial APSR (↓)** | 68.82↑63% | 68.32↑63% | 69.48↑64% | 69.02↑63% | 65.92↑60% | 71.57↑65% | 70.47↑65% | 69.12↑64% |

Table 5: Performance metrics (IOU and APSR) for various self-supervised and supervised models under different adversarial attacks, using frozen backbones. Clean and adversarial scores are reported, with percentage changes in adversarial performance noted. Higher IOU and lower APSR indicate better results.

B.3 DETECTION

Table 6: Adversarial Attack Results on Detection using Unfrozen SSL and Supervised Models as backbones. The table presents performance metrics under clean and adversarial conditions for various attack types (Optim, BIM, MIM, SGD, PGD, Optim-Adam, Optim-Nesterov). The last two rows display clean mean Average Precision (mAP) and the average performance under adversarial attacks, with the percentage decrease in performance highlighted in red

|  | Barlow | BYOL | DINO | MocoV3 | SimCLR | Supervised | SwAV | VICReg |
|---|---|---|---|---|---|---|---|---|
| Clean | 89.14 | 88.98 | 89.74 | 89.74 | 89.01 | 86.45 | 88.60 | 89.45 |
| Optim | 6.18 | 1.68 | 1.77 | 4.87 | 2.11 | 1.54 | 4.27 | 2.12 |
| BIM | 32.78 | 26.93 | 31.82 | 21.63 | 13.62 | 1.75 | 40.84 | 23.22 |
| MIM | 11.89 | 26.24 | 5.2 | 10.7 | 5.38 | 1.94 | 10.69 | 7.85 |
| SGD | 6.13 | 2.89 | 7.59 | 20.15 | 12.58 | 2.4 | 13.71 | 2.99 |
| PGD | 84.58 | 78.44 | 80.97 | 81.96 | 80.88 | 57.76 | 80.54 | 77.52 |
| Optim-Adam | 6.43 | 1.49 | 2.07 | 7.49 | 2.18 | 1.32 | 4.47 | 1.99 |
| Optim-Nesterov | 2.34 | 1.58 | 1.31 | 5.24 | 1.93 | 2.55 | 4.34 | 1.42 |
| Clean mAP | 89.14 | 88.98 | 89.74 | 89.74 | 89.01 | 86.45 | 88.60 | 89.45 |
| Adv Avg. | 21.48↓76% | 19.89↓78% | 18.68↓79% | 21.72↓76% | 16.95↓81% | 9.89↓89% | 22.69↓72% | 16.73↓81% |

Table 7: Adversarial Attack Results on Detection using frozen SSL and Supervised Models as backbones. The table presents performance metrics under clean and adversarial conditions for various attack types (Optim, BIM, MIM, SGD, PGD, Optim-Adam, Optim-Nesterov). The last two rows display clean mean Average Precision (mAP) and the average performance under adversarial attacks, with the percentage decrease in performance highlighted in red

|  | Barlow | BYOL | DINO | MocoV3 | SimCLR | Supervised | SwAV | VICReg |
|---|---|---|---|---|---|---|---|---|
| Optim | 3.98 | 1.05 | 2 | 2.6 | 0.65 | 0.56 | 1.51 | 0.39 |
| BIM | 44.87 | 32.24 | 54.93 | 26.72 | 17.1 | 42.8 | 44.47 | 10.32 |
| MIM | 11.37 | 3.04 | 10.32 | 5.72 | 7.45 | 4.73 | 10.87 | 2.68 |
| SGD | 3.21 | 1.28 | 2.95 | 9.44 | 4.3 | 1.02 | 2.85 | 1.72 |
| PGD | 83.08 | 80.83 | 79.65 | 79.83 | 76.9 | 75.29 | 79.14 | 81.27 |
| Optim-Adam | 4.71 | 0.76 | 3.5 | 2.03 | 0.81 | 0.87 | 3.46 | 0.67 |
| Optim-Nesterov | 1.75 | 0.64 | 0.97 | 2.77 | 0.62 | 0.72 | 1.1 | 0.64 |
| Clean mAP | 88.39 | 87.44 | 87.63 | 87.36 | 88.27 | 86.08 | 86.55 | 88.43 |
| Adv Avg. | 21.85↓75% | 17.12↓80% | 22.05↓75% | 18.44↓79% | 15.40↓83% | 18.00↓79% | 20.49↓76% | 13.96↓84% |

## B.4 RESNET VS VIT

Table 8: This table presents the results of various instance and universal adversarial perturbation (UAP) attacks on the Imagenet-1k dataset, with all UAP attack names in *italics*. Different configurations of FGSM and PGD are denoted, such as $FGSM_1$ and $PGD_1$. Average results for universal adversarial perturbations (UAP Avg.), instance adversarial attacks (IAA Avg.), and overall adversarial performance (Adv Avg.) are reported at the bottom, including percentage drops relative to clean accuracy

|  | MoCoV3-ResNet | MoCo-ViT | DINO-ViT | DINO-ResNet |
|---|---|---|---|---|
| $FGSM_1$ | 42.67 | 34.63 | 51.42 | 24.68 |
| $FGSM_2$ | 15.53 | 0.32 | 0.97 | 5.66 |
| $PGD_1$ | 42.39 | 33.35 | 50.98 | 25.65 |
| $PGD_2$ | 1.06 | 0.00 | 0.00 | 0.18 |
| $PGD_3$ | 42.56 | 33.46 | 50.95 | 25.85 |
| $PGD_4$ | 0.82 | 0.17 | 3.84 | 0.14 |
| $PGD_5$ | 4.79 | 2.12 | 13.57 | 0.67 |
| DIFGSM | 54.09 | 51.91 | 59.81 | 41.12 |
| CW | 0.02 | 0 | 0 | 0 |
| Jitter | 62.47 | 58.25 | 66.30 | 60.26 |
| TIFGSM | 61.47 | 61.84 | 65.23 | 56.98 |
| PIFGSM | 34.10 | 25.78 | 47.64 | 14.54 |
| EADEN | 0 | 0 | 0 | 0 |
| OnePixel | 72.64 | 71.28 | 75.47 | 72.83 |
| Pixle | 31.45 | 34.69 | 44.08 | 19.41 |
| SPSA | 69.93 | 66.20 | 72.47 | 68.11 |
| Square | 3.15 | 1.22 | 1.67 | 1.30 |
| TAP | 73.72 | 72.34 | 75.60 | 73.78 |
| *ASV* | 50.21 | 46.28 | 48.1 | 45.08 |
| *FFF (no-data)* | 49.63 | 46.49 | 50.41 | 43.58 |
| *FFF (mean-std)* | 49.01 | 48.46 | 50.31 | 43.58 |
| *FFF (one-sample)* | 49.9 | 48.47 | 50.02 | 44.5 |
| *FG-UAP* | 45.28 | 34.95 | 41.58 | 37.41 |
| *GD-UAP (no-data)* | 49.41 | 46.97 | 48.86 | 43.71 |
| *GD-UAP (mean-std)* | 49.33 | 46.04 | 48.39 | 43.78 |
| *GD-UAP (one-sample)* | 49.98 | 46.62 | 48.41 | 44.59 |
| *L4A-base* | 49.51 | 33.59 | 44.38 | 44.61 |
| *L4A-fuse* | 49.48 | 34.59 | 44.39 | 44.64 |
| *L4A-ugs* | 49.83 | 37.32 | 45.1 | 45.25 |
| *PD-UAP* | 50.05 | 46.81 | 50.7 | 44.14 |
| *SSP* | 47.61 | 32.43 | 43.59 | 43.09 |
| *STD* | 49.4 | 46.8 | 48.98 | 44.16 |
| *UAP (DeepFool)* | 50.43 | 43.81 | 48.55 | 45.43 |
| *UAPEPGD* | 50.67 | 47.98 | 50.49 | 45.54 |
| Clean Accuracy | 74.57 | 73.21 | 76.95 | 75.28 |
| IAA Avg. | 34.05 ↓54% | 30.42↓58% | 37.78↓51% | 28.29↓64% |
| UAP Avg. | 49.36 ↓34% | 42.97↓41% | 47.64↓38% | 43.94↓42% |
| Adv Avg. | 41.26 ↓45% | 36.32↓50% | 42.42↓45% | 35.13↓53% |

## B.5 IMAGENET ACROSS TRAINING EPOCHS

Table 9: This table presents the results of various instance and universal adversarial perturbation (UAP) attacks on the Imagenet-1k dataset, with all UAP attack names in *italics*. Different configurations of FGSM and PGD are denoted, such as $FGSM_1$ and $PGD_1$. Average results for universal adversarial perturbations (UAP Avg.), instance adversarial attacks (IAA Avg.), and overall adversarial performance (Adv Avg.) are reported at the bottom, including percentage drops relative to clean accuracy.

|  | MoCoV3-100 | MoCoV3-300 | MoCoV3-1000 |
|---|---|---|---|
| $FGSM_1$ | 38.87 | 42.6 | 42.67 |
| $FGSM_2$ | 7.94 | 8.38 | 15.53 |
| $PGD_1$ | 37.89 | 41.99 | 42.39 |
| $PGD_2$ | 0.49 | 0.09 | 1.06 |
| $PGD_3$ | 38.06 | 42.14 | 42.56 |
| $PGD_4$ | 1.75 | 1.22 | 0.82 |
| $PGD_5$ | 5.4 | 5.49 | 4.79 |
| DIFGSM | 49.21 | 52.65 | 54.09 |
| CW | 0.02 | 0.02 | 0.02 |
| Jitter | 56.45 | 60.53 | 62.47 |
| TIFGSM | 57.39 | 61.86 | 61.47 |
| PIFGSM | 31.24 | 34.41 | 34.1 |
| EADEN | 0 | 0 | 0 |
| OnePixel | 66.79 | 70.76 | 72.64 |
| Pixle | 26.27 | 29.41 | 31.45 |
| SPSA | 64.05 | 68.02 | 69.93 |
| Square | 2.05 | 2.01 | 3.15 |
| TAP | 67.85 | 71.9 | 73.72 |
| *ASV* | 43.31 | 48.69 | 50.21 |
| *FFF (no-data)* | 42.65 | 47.68 | 49.63 |
| *FFF (mean-std)* | 42.51 | 48.13 | 49.01 |
| *FFF (one-sample)* | 42.9 | 48.33 | 49.9 |
| *FG-UAP* | 39.68 | 44.49 | 45.28 |
| *GD-UAP (no-data)* | 42.53 | 47.99 | 49.41 |
| *GD-UAP (mean-std)* | 42.59 | 48.09 | 49.33 |
| *GD-UAP (one-sample)* | 42.93 | 48.38 | 49.98 |
| *L4A-base* | 41.95 | 48.74 | 49.51 |
| *L4A-fuse* | 41.96 | 48.82 | 49.48 |
| *L4A-ugs* | 43.12 | 48.87 | 49.83 |
| *PD-UAP* | 43.21 | 48.46 | 50.05 |
| *SSP* | 42.53 | 46.5 | 47.61 |
| *STD* | 42.34 | 47.97 | 49.4 |
| *UAP (DeepFool)* | 43.39 | 48.95 | 50.43 |
| *UAPEPGD* | 43.73 | 49.22 | 50.67 |
| Clean Accuracy | 68.91 | 72.82 | 74.57 |
| IAA Avg. | 30.65 ↓56% | 32.97 ↓55% | 34.05 ↓54% |
| UAP Avg. | 42.58 ↓38% | 48.08 ↓34% | 49.36 ↓34% |
| Adv Avg. | 36.26 ↓47% | 40.08 ↓45% | 41.26 ↓45% |

Table 10: This table presents the results of various instance and universal adversarial perturbation (UAP) attacks on the Imagenet-1k dataset, with all UAP attack names in *italics*. Different configurations of FGSM and PGD are denoted, such as $FGSM_1$ and $PGD_1$. Average results for universal adversarial perturbations (UAP Avg.), instance adversarial attacks (IAA Avg.), and overall adversarial performance (Adv Avg.) are reported at the bottom, including percentage drops relative to clean accuracy

|  | SwAV-100 | SwAV-200 | SwAV-400 | SwAV-800 |
|---|---|---|---|---|
| $FGSM_1$ | 18.08 | 19.99 | 21.9 | 24.71 |
| $FGSM_2$ | 4.01 | 4.34 | 5.2 | 6.35 |
| $PGD_1$ | 18.94 | 21.3 | 23.7 | 26.48 |
| $PGD_2$ | 0.31 | 0.17 | 0.17 | 0.18 |
| $PGD_3$ | 19.08 | 21.44 | 23.88 | 26.73 |
| $PGD_4$ | 0.3 | 0.15 | 0.14 | 0.14 |
| $PGD_5$ | 0.73 | 0.59 | 0.52 | 0.69 |
| DIFGSM | 39.31 | 42.01 | 42.31 | 45.65 |
| CW | 0.0 | 0.0 | 0.0 | 0.0 |
| Jitter | 56.67 | 59.15 | 60.43 | 61.16 |
| TIFGSM | 53.11 | 55.14 | 56.44 | 60.07 |
| PIFGSM | 10 | 10.87 | 11.76 | 14.12 |
| EADEN | 0 | 0 | 0 | 0 |
| OnePixel | 68.73 | 70.83 | 71.64 | 72.73 |
| Pixle | 13.21 | 16.03 | 18.08 | 16.95 |
| SPSA | 63.94 | 66.25 | 67.38 | 68.61 |
| Square | 0.35 | 0.36 | 0.5 | 1.99 |
| TAP | 71.79 | 73.56 | 74.37 | 75.05 |
| *ASV* | 47.64 | 50.84 | 52.32 | 53.66 |
| *FFF (no-data)* | 45.54 | 49.34 | 49.99 | 51.88 |
| *FFF (mean-std)* | 46.65 | 50.38 | 50.26 | 53.4 |
| *FFF (one-sample)* | 46.42 | 50.28 | 51.13 | 53.33 |
| *FG-UAP* | 36.34 | 40.47 | 42.19 | 44.59 |
| *GD-UAP (no-data)* | 45.56 | 49.39 | 50.46 | 52.19 |
| *GD-UAP (mean-std)* | 46.54 | 50.26 | 50.85 | 53.11 |
| *GD-UAP (one-sample)* | 46.63 | 50.34 | 51.32 | 53.4 |
| *L4A-base* | 44.01 | 48.94 | 49.18 | 49.89 |
| *L4A-fuse* | 43.86 | 48.88 | 49 | 49.69 |
| *L4A-ugs* | 44.25 | 49.94 | 50.9 | 53.37 |
| *PD-UAP* | 45.86 | 49.68 | 51.37 | 53.45 |
| *SSP* | 42.12 | 48.29 | 47.25 | 51.21 |
| *STD* | 46.6 | 50.3 | 51.41 | 53.18 |
| *UAP (DeepFool)* | 46.86 | 50.77 | 51.77 | 53.86 |
| *UAPEPGD* | 47.82 | 51.32 | 52.31 | 54.26 |
| Clean Accuracy | 72.02 | 73.82 | 74.57 | 75.27 |
| IAA Avg. | 24.36 ↓66% | 25.68↓65% | 26.58↓64% | 27.87↓63% |
| UAP Avg. | 45.17 ↓37% | 49.34↓33% | 50.11↓33% | 52.15↓31% |
| Adv Avg. | 34.15 ↓53% | 36.81↓50% | 37.65↓49% | 39.29↓48% |

## B.6 IMAGENET WITH DIFFERENT MOCO VERSIONS

Table 11: This table presents the results of various instance and universal adversarial perturbation (UAP) attacks on the Imagenet-1k dataset, with all UAP attack names in *italics*. Different configurations of FGSM and PGD are denoted, such as $FGSM_1$ and $PGD_1$. Average results for universal adversarial perturbations (UAP Avg.), instance adversarial attacks (IAA Avg.), and overall adversarial performance (Adv Avg.) are reported at the bottom, including percentage drops relative to clean accuracy

| | MoCoV1 | MoCoV2 | MoCoV3 |
|---|---|---|---|
| $FGSM_1$ | 15.91 | 22.01 | 42.67 |
| $FGSM_2$ | 6.25 | 5.17 | 15.53 |
| $PGD_1$ | 17.89 | 24.00 | 42.39 |
| $PGD_2$ | 0.09 | 0.54 | 1.06 |
| $PGD_3$ | 17.96 | 24.14 | 42.56 |
| $PGD_4$ | 0.06 | 0.52 | 0.82 |
| $PGD_5$ | 0.21 | 1.33 | 4.79 |
| DIFGSM | 34.85 | 40.39 | 54.09 |
| CW | 0 | 0 | 0.02 |
| Jitter | 50.04 | 53.09 | 62.47 |
| TIFGSM | 48.70 | 49.50 | 61.47 |
| PIFGSM | 8.53 | 13.20 | 34.10 |
| EADEN | 0 | 0 | 0 |
| OnePixel | 56.67 | 64.63 | 72.64 |
| Pixle | 3.10 | 17.85 | 31.45 |
| SPSA | 50.62 | 60.57 | 69.93 |
| Square | 0.80 | 0.42 | 3.15 |
| TAP | 58.55 | 65.24 | 73.72 |
| *ASV* | 19.18 | 40.17 | 50.21 |
| *FFF (no-data)* | 23.41 | 39.43 | 49.63 |
| *FFF (mean-std)* | 23.89 | 39.35 | 49.01 |
| *FFF (one-sample)* | 23.49 | 39.47 | 49.9 |
| *FG-UAP* | 13.25 | 35.73 | 45.28 |
| *GD-UAP (no-data)* | 23.72 | 39.79 | 49.41 |
| *GD-UAP (mean-std)* | 24.07 | 39.47 | 49.33 |
| *GD-UAP (one-sample)* | 23.64 | 39.73 | 49.98 |
| *L4A-base* | 12.25 | 39.76 | 49.51 |
| *L4A-fuse* | 12.14 | 39.6 | 49.48 |
| *L4A-ugs* | 12.43 | 39.96 | 49.83 |
| *PD-UAP* | 23.27 | 40.24 | 50.05 |
| *SSP* | 12.49 | 39.01 | 47.61 |
| *STD* | 24.32 | 39.55 | 49.4 |
| *UAP (DeepFool)* | 18.43 | 40.42 | 50.43 |
| *UAPEPGD* | 26.08 | 40.71 | 50.67 |
| Clean Accuracy | 60.64 | 67.72 | 74.57 |
| IAA Avg. | 20.56 ↓66% | 24.59↓64% | 34.05↓54% |
| UAP Avg. | 19.75 ↓67% | 39.52↓42% | 49.36↓34% |
| Adv Avg. | 20.19 ↓67% | 31.61↓53% | 41.26↓45% |

## B.7 TRANSFER LEARNING

Table 12: This table presents the combined results from each transfer learning dataset. Average results for universal adversarial perturbations (UAP Avg.), instance adversarial attacks (IAA Avg.), and overall adversarial performance (Adv Avg.) are reported at the bottom, including percentage drops relative to clean accuracy

| | | Barlow | BYOL | DINO | MocoV3 | SimCLR | Supervised | SwAV | VICReg |
|---|---|---|---|---|---|---|---|---|---|
| Aircraft | Clean Accuracy | 56.88 | 56.34 | 60.25 | 58.75 | 46.77 | 44.89 | 54.01 | 56.43 |
| | IAA Avg. | 16.29↓71% | 14.87↓73% | 15.27↓75% | 17.41↓70% | 11.93↓74% | 9.82↓78% | 13.82↓74% | 16.38↓71% |
| | UAP Avg. | 24.48↓57% | 35.94↓36% | 20.62↓66% | 27.02↓54% | 13.4↓72% | 10.75↓77% | 20.16↓63% | 24.42↓57% |
| | Adv Avg. | 20.14↓65% | 24.78↓56% | 17.78↓70% | 21.93↓63% | 12.62↓73% | 10.25↓77% | 16.80↓69% | 20.16↓64% |
| Caltech 101 | Clean Accuracy | 90.54 | 90.99 | 90.31 | 92.89 | 89.1 | 90.25 | 90.36 | 90.57 |
| | IAA Avg. | 53.60↓41% | 54.06↓41% | 47.42↓47% | 58.23↓37% | 49.79↓44% | 44.10↓51% | 45.55↓50% | 53.64↓41% |
| | UAP Avg. | 71.86↓21% | 82.04↓10% | 61.70↓32% | 80.95↓13% | 67.06↓25% | 58.86↓35% | 74.36↓17.7% | 71.83↓21% |
| | Adv Avg. | 62.19↓31% | 67.22↓26% | 54.14↓40% | 68.92↓26% | 57.92↓35% | 51.05↓43% | 59.11↓35% | 62.20↓31% |
| Cars | Clean Accuracy | 64.2 | 57.62 | 65.62 | 63.61 | 43.81 | 47.1 | 59.78 | 64.12 |
| | IAA Avg. | 19.90↓69% | 15.84↓73% | 17.54↓73% | 20.12↓68% | 11.14↓75% | 9.56↓80% | 14.95↓75% | 19.66↓69% |
| | UAP Avg. | 26.89↓58% | 36.71↓36% | 22.45↓66% | 32.82↓48% | 18.07↓59% | 9.27↓80% | 24.43↓59% | 26.52↓59% |
| | Adv Avg. | 23.19↓64% | 25.66↓55% | 19.85↓70% | 26.09↓60% | 14.40↓67% | 9.42↓80% | 19.41↓68% | 22.89↓64% |
| CIFAR 10 | Clean Accuracy | 92.78 | 93.05 | 93.85 | 94.67 | 90.98 | 91.4 | 93.9 | 92.79 |
| | IAA Avg. | 32.34↓65% | 31.19↓66% | 28.07↓70% | 32.85↓65% | 30.00↓67% | 31.74↓65% | 27.37↓71% | 32.45↓65% |
| | UAP Avg. | 43.68↓53% | 51.76↓44% | 32.78↓65% | 41.92↓56% | 25.28↓72% | 29.27↓68% | 33.84↓64% | 43.91↓53% |
| | Adv Avg. | 37.68↓59% | 40.87↓56% | 30.28↓68% | 37.12↓61% | 27.78↓69% | 30.58↓66% | 30.41↓68% | 37.84↓59% |
| CIFAR 100 | Clean Accuracy | 77.86 | 78.18 | 76.67 | 80.19 | 72.97 | 73.86 | 79.41 | 77.79 |
| | IAA Avg. | 23.34↓70% | 22.65↓71% | 20.45↓74% | 22.77↓72% | 18.36↓75% | 21.72↓71% | 19.59↓75% | 24.05↓69% |
| | UAP Avg. | 24.86↓68% | 35.15↓55% | 16.52↓79% | 21.89↓73% | 10.33↓86% | 14.56↓80% | 21.18↓73% | 25.70↓67% |
| | Adv Avg. | 24.06↓69% | 28.53↓63% | 18.55↓77% | 22.36↓72% | 14.58↓80% | 18.34↓75% | 20.34↓74% | 24.82↓68% |
| DTD | Clean Accuracy | 79.97 | 76.76 | 77.02 | 75.43 | 73.19 | 72.13 | 77.45 | 77.61 |
| | IAA Avg. | 40.02↓50% | 37.65↓51% | 38.88↓50% | 40.14↓50% | 33.50↓54% | 33.86↓53% | 38.96↓50% | 41.30↓47% |
| | UAP Avg. | 52.85↓34% | 61.65↓17% | 48.88↓37% | 56.44↓25% | 52.96↓28% | 38.44↓47% | 57.26↓26% | 53.78↓31% |
| | Adv Avg. | 46.06↓42% | 48.94↓34% | 43.58↓43% | 47.81↓37% | 42.66↓42% | 36.02↓50% | 47.57↓39% | 47.17↓39% |
| Flowers | Clean Accuracy | 94.92 | 93.36 | 95.23 | 94.07 | 90.57 | 90.59 | 93.84 | 94.92 |
| | IAA Avg. | 47.71↓50% | 43.94↓53% | 43.76↓54% | 47.25↓50% | 40.25↓56% | 34.86↓62% | 39.92↓58% | 47.94↓50% |
| | UAP Avg. | 74.25↓22% | 81.84↓12% | 68.05↓29% | 74.97↓20% | 56.01↓38% | 33.83↓63% | 70.01↓25% | 74.20↓22% |
| | Adv Avg. | 60.19↓37% | 61.78↓34% | 55.19↓42% | 60.30↓36% | 47.66↓47% | 34.37↓62% | 54.08↓42% | 60.30↓37% |
| Food | Clean Accuracy | 76.09 | 73.07 | 78.42 | 73.83 | 67.24 | 69.05 | 76.51 | 75.81 |
| | IAA Avg. | 27.50↓64% | 24.15↓67% | 24.09↓69% | 27.69↓62% | 21.03↓69% | 19.81↓71% | 23.39↓69% | 26.37↓65% |
| | UAP Avg. | 40.04↓47% | 48.81↓33% | 38.41↓51% | 43.09↓42% | 32.94↓51% | 19.36↓72% | 43.73↓43% | 39.03↓49% |
| | Adv Avg. | 33.40↓56% | 35.75↓51% | 30.83↓61% | 34.94↓53% | 26.63↓60% | 19.59↓72% | 32.96↓57% | 32.33↓57% |
| Pets | Clean Accuracy | 89.13 | 89.08 | 89.15 | 90.77 | 83.23 | 92.06 | 87.47 | 89.13 |
| | IAA Avg. | 45.87↓49% | 44.48↓50% | 39.48↓56% | 50.74↓44% | 37.75↓55% | 41.79↓55% | 36.73↓58% | 45.95↓48.4% |
| | UAP Avg. | 63.22↓29% | 75.21↓16% | 62.43↓30% | 69.77↓23% | 61.16↓27% | 49.33↓46% | 65.30↓25% | 63.22↓29% |
| | Adv Avg. | 54.03↓39% | 58.94↓34% | 50.28↓44% | 59.69↓34% | 48.77↓41% | 45.34↓51% | 50.18↓426% | 54.08↓39% |
| *All* | *Clean Accuracy* | 80.26 | 78.71 | 80.72 | 80.47 | 73.09 | 74.59 | 79.19 | 79.90 |
| | *IAA Avg.* | 34.06↓58% | 32.09↓59% | 30.55↓62% | 35.24↓56% | 28.19↓62% | 27.47↓63% | 28.92↓63% | 34.19↓57% |
| | *UAP Avg.* | 46.90↓42% | 64.36↓18% | 41.31↓49% | 49.87↓38% | 37.46↓49% | 25.53↓66% | 45.58↓42% | 46.95↓41% |
| | *Adv Avg.* | 40.10↓50% | 47.27↓40% | 35.61↓56% | 42.12↓47% | 32.55↓55% | 26.55↓64% | 35.66↓55% | 39.86↓50% |

### B.7.1 AirCraft

Table 13: This table presents the results of various instance and universal adversarial perturbation (UAP) attacks on the AirCraft dataset, with all UAP attack names in *italics*. Different configurations of FGSM and PGD are denoted, such as $FGSM_1$ and $PGD_1$. Average results for universal adversarial perturbations (UAP Avg.), instance adversarial attacks (IAA Avg.), and overall adversarial performance (Adv Avg.) are reported at the bottom, including percentage drops relative to clean accuracy.

| | Barlow | BYOL | DINO | MoCoV3 | SimCLR | Supervised | SwAV | VICReg |
|---|---|---|---|---|---|---|---|---|
| $FGSM_1$ | 8.92 | 5.94 | 4.84 | 11.41 | 2.7 | 2.58 | 3.64 | 8.86 |
| $FGSM_2$ | 1.52 | 0.69 | 0.45 | 1.95 | 0.78 | 0.81 | 2.57 | 1.8 |
| $PGD_1$ | 10.03 | 5.72 | 4.54 | 10.96 | 3.44 | 1.61 | 4 | 10.18 |
| $PGD_2$ | 0.06 | 0 | 0 | 0.12 | 0.24 | 0.18 | 0.64 | 0.06 |
| $PGD_3$ | 10.27 | 6.02 | 4.63 | 11.09 | 3.27 | 1.61 | 3.83 | 10.06 |
| $PGD_4$ | 0.06 | 0 | 0 | 0.12 | 0.18 | 0.12 | 0.61 | 0.06 |
| $PGD_5$ | 0.12 | 0.03 | 0 | 0.24 | 0.18 | 0.24 | 0.79 | 0.12 |
| DIFGSM | 24.56 | 24.16 | 20.83 | 28.01 | 19.39 | 19.43 | 16.74 | 27.41 |
| CW | 0 | 0 | 0 | 0 | 0 | 0 | 0 | 0 |
| Jitter | 45.87 | 44.28 | 48.39 | 45.42 | 37.43 | 31.98 | 43.75 | 44.73 |
| TIFGSM | 32.78 | 31.08 | 29.68 | 35.76 | 28.31 | 18.99 | 29.83 | 33.04 |
| PIFGSM | 3.62 | 2.1 | 1.62 | 4.46 | 0.9 | 0.6 | 1.71 | 3.44 |
| EADEN | 0 | 0 | 0 | 0 | 0 | 0 | 0 | 0 |
| OnePixel | 51.75 | 49.39 | 54.93 | 53.41 | 41.4 | 36.01 | 47.55 | 51.54 |
| Pixle | 3.67 | 1.9 | 2.17 | 6.16 | 2.8 | 1.48 | 2.26 | 3.8 |
| SPSA | 44.36 | 42.91 | 44.2 | 46.6 | 30.76 | 28.51 | 38.42 | 44.31 |
| Square | 0.03 | 0 | 0 | 0.03 | 0.03 | 0 | 0 | 0.03 |
| TAP | 55.53 | 53.4 | 58.55 | 57.72 | 42.93 | 32.54 | 52.48 | 55.35 |
| *ASV* | 25.95 | 38.29 | 24.26 | 31.96 | 22.42 | 10.93 | 23.25 | 25.64 |
| *FFF (no-data)* | 23.58 | 34.84 | 17.23 | 26.2 | 14.04 | 9.95 | 17.14 | 23.64 |
| *FFF (mean-std)* | 23.76 | 34.28 | 18.44 | 23.37 | 13.89 | 10.76 | 21.86 | 23.59 |
| *FFF (one-sample)* | 26.01 | 35.97 | 21.68 | 28.8 | 13.57 | 10.82 | 20.47 | 25.52 |
| *FG-UAP* | 14.34 | 31.84 | 13.49 | 17.41 | 4.07 | 8.22 | 11.84 | 14.58 |
| *GD-UAP (no-data)* | 24.39 | 35.92 | 17.62 | 24.93 | 17.55 | 10.22 | 17.19 | 23.63 |
| *GD-UAP (mean-std)* | 24.3 | 32.51 | 19.34 | 23.24 | 13.46 | 11.81 | 20.42 | 24.88 |
| *GD-UAP (one-sample)* | 25.95 | 36.04 | 22.07 | 28.36 | 14.87 | 10.79 | 20.28 | 26.36 |
| *L4A-base* | 25.89 | 35.17 | 21.29 | 26.91 | 4.29 | 11.42 | 18.2 | 25.77 |
| *L4A-fuse* | 25.95 | 35.18 | 20.65 | 26.91 | 4.07 | 11.42 | 18.14 | 25.62 |
| *L4A-ugs* | 26.02 | 38.54 | 23.99 | 29.73 | 24.55 | 10.97 | 22.41 | 26.26 |
| *PD-UAP* | 24 | 37.7 | 18.14 | 28.96 | 16.65 | 10.4 | 21.16 | 24.24 |
| *SSP* | 20.25 | 33.19 | 18.8 | 22.7 | 16.57 | 9.79 | 19.87 | 20.09 |
| *STD* | 26.2 | 36.67 | 23.54 | 28.41 | 9.08 | 10.61 | 21.39 | 25.68 |
| *UAP (DeepFool)* | 26.78 | 39.06 | 24.19 | 31.26 | 5.33 | 11.83 | 23.27 | 26.9 |
| *UAPEPGD* | 28.34 | 39.82 | 25.21 | 33.21 | 19.95 | 12.13 | 25.65 | 28.25 |
| Clean Accuracy | 56.88 | 56.34 | 60.25 | 58.75 | 46.77 | 44.89 | 54.01 | 56.43 |
| IAA Avg. | 16.29 ↓71% | 14.87↓73% | 15.27↓75% | 17.41↓70% | 11.93↓74% | 9.82↓78% | 13.82↓74% | 16.38↓71% |
| UAP Avg. | 24.48 ↓57% | 35.94↓36% | 20.62↓66% | 27.02↓54% | 13.4↓72% | 10.75↓77% | 20.16↓63% | 24.42↓57% |
| Adv Avg. | 20.14 ↓65% | 24.78↓56% | 17.78↓70% | 21.93↓63% | 12.62↓73% | 10.25↓77% | 16.80↓69% | 20.16↓64% |

### B.7.2 CALTECH 101

Table 14: This table presents the results of various instance and universal adversarial perturbation (UAP) attacks on the Caltech 101 dataset, with all UAP attack names in *italics*. Different configurations of FGSM and PGD are denoted, such as $FGSM_1$ and $PGD_1$. Average results for universal adversarial perturbations (UAP Avg.), instance adversarial attacks (IAA Avg.), and overall adversarial performance (Adv Avg.) are reported at the bottom, including percentage drops relative to clean accuracy.

| | Barlow | BYOL | DINO | MoCoV3 | SimCLR | Supervised | SwAV | VICReg |
|---|---|---|---|---|---|---|---|---|
| $FGSM_1$ | 75.31 | 75.58 | 66.93 | 79.84 | 66.06 | 62.11 | 63.12 | 75.3 |
| $FGSM_2$ | 53.82 | 52.44 | 37.84 | 59.58 | 47.67 | 27.38 | 36.13 | 53.82 |
| $PGD_1$ | 74.27 | 75.19 | 65.57 | 79.35 | 64.94 | 58.96 | 61.96 | 74.34 |
| $PGD_2$ | 9.61 | 10.47 | 2.24 | 17.17 | 11.14 | 1.64 | 2.05 | 9.34 |
| $PGD_3$ | 74.43 | 75.39 | 65.7 | 79.81 | 65 | 59.24 | 62.28 | 74.68 |
| $PGD_4$ | 7.62 | 9 | 1.81 | 14.79 | 10.22 | 1.19 | 1.69 | 7.53 |
| $PGD_5$ | 17.17 | 18.64 | 5.48 | 25.45 | 13.11 | 4.35 | 3.91 | 16.86 |
| DIFGSM | 80.24 | 81.09 | 76.38 | 83.66 | 76.16 | 71.28 | 75.23 | 79.97 |
| CW | 0.68 | 0.94 | 0.3 | 0.79 | 0.49 | 0.22 | 0.31 | 0.68 |
| Jitter | 83.43 | 83.41 | 81.7 | 86.82 | 80.89 | 77.36 | 79.34 | 83.85 |
| TIFGSM | 85.73 | 86.72 | 83.63 | 88.69 | 82.73 | 79.58 | 81.98 | 85.98 |
| PIFGSM | 68.03 | 68.03 | 53.66 | 74.14 | 50.54 | 49 | 45.82 | 67.98 |
| EADEN | 0 | 0 | 0 | 0 | 0 | 0 | 0 | 0 |
| OnePixel | 89.85 | 90.57 | 89.43 | 92.25 | 87.67 | 88.7 | 89.52 | 89.88 |
| Pixle | 53.89 | 57.26 | 40.6 | 67.39 | 49.57 | 39.02 | 32.73 | 54.58 |
| SPSA | 88.89 | 88.82 | 87.45 | 91.08 | 86.51 | 85.73 | 87.2 | 89.04 |
| Square | 11.43 | 8.71 | 4.7 | 14.98 | 15.14 | 1.03 | 6.53 | 11.37 |
| TAP | 90.48 | 90.91 | 90.16 | 92.36 | 88.52 | 87.13 | 90.12 | 90.48 |
| *ASV* | 71.34 | 81.96 | 62.14 | 81.76 | 85.97 | 59.01 | 75.6 | 71.59 |
| *FFF (no-data)* | 72.78 | 82.35 | 61.23 | 81.19 | 64.38 | 59.04 | 74.38 | 72.22 |
| *FFF (mean-std)* | 72.02 | 82.09 | 61.7 | 80.87 | 73.87 | 59.55 | 75.57 | 72.16 |
| *FFF (one-sample)* | 72.38 | 81.76 | 62.8 | 81.24 | 69.45 | 58.79 | 75.78 | 72.31 |
| *FG-UAP* | 69.93 | 81.14 | 54.08 | 77.94 | 14.44 | 55.00 | 66.22 | 70.01 |
| *GD-UAP (no-data)* | 72.37 | 82.21 | 61.26 | 80.97 | 74.03 | 58.81 | 74.81 | 72.30 |
| *GD-UAP (mean-std)* | 72.25 | 81.87 | 62.04 | 80.72 | 80.36 | 59.04 | 74.91 | 71.85 |
| *GD-UAP (one-sample)* | 72.06 | 82.04 | 62.31 | 81.64 | 73.19 | 59.00 | 75.88 | 72.08 |
| *L4A-base* | 71.62 | 82.03 | 63.02 | 80.93 | 37.65 | 59.02 | 71.93 | 71.42 |
| *L4A-fuse* | 71.41 | 81.78 | 63.07 | 80.98 | 37.32 | 59.11 | 71.08 | 71.32 |
| *L4A-ugs* | 72.16 | 82.48 | 62.85 | 81.49 | 81.71 | 58.88 | 75.75 | 72.16 |
| *PD-UAP* | 72.89 | 82.08 | 62.20 | 81.35 | 84.24 | 59.48 | 75.86 | 72.70 |
| *SSP* | 70.20 | 81.98 | 60.70 | 79.76 | 76.76 | 58.27 | 73.95 | 70.45 |
| *STD* | 71.87 | 82.34 | 62.47 | 81.30 | 81.32 | 59.20 | 76.07 | 72.09 |
| *UAP (DeepFool)* | 72.07 | 82.22 | 62.81 | 81.44 | 52.16 | 59.97 | 75.84 | 72.28 |
| *UAPEPGD* | 72.47 | 82.31 | 62.66 | 81.66 | 86.14 | 59.69 | 76.23 | 72.35 |
| Clean Accuracy | 90.54 | 90.99 | 90.31 | 92.89 | 89.1 | 90.25 | 90.36 | 90.57 |
| IAA Avg. | 53.60 ↓41% | 54.06 ↓41% | 47.42 ↓47% | 58.23 ↓37% | 49.79 ↓44% | 44.10 ↓51% | 45.55 ↓50% | 53.64 ↓41% |
| UAP Avg. | 71.86 ↓21% | 82.04 ↓10% | 61.70 ↓32% | 80.95 ↓13% | 67.06 ↓25% | 58.86 ↓35% | 74.36 ↓17.7% | 71.83 ↓21% |
| Adv Avg. | 62.19 ↓31% | 67.22 ↓26% | 54.14 ↓40% | 68.92 ↓26% | 57.92 ↓35% | 51.05 ↓43% | 59.11 ↓35% | 62.20 ↓31% |

### B.7.3 Cars

Table 15: This table presents the results of various instance and universal adversarial perturbation (UAP) attacks on the Cars dataset, with all UAP attack names in *italics*. Different configurations of FGSM and PGD are denoted, such as $FGSM_1$ and $PGD_1$. Average results for universal adversarial perturbations (UAP Avg.), instance adversarial attacks (IAA Avg.), and overall adversarial performance (Adv Avg.) are reported at the bottom, including percentage drops relative to clean accuracy.

| | Barlow | BYOL | DINO | MoCoV3 | SimCLR | Supervised | SwAV | VICReg |
|---|---|---|---|---|---|---|---|---|
| $FGSM_1$ | 14.55 | 8.27 | 6.34 | 16.32 | 3.18 | 2.1 | 3.48 | 14.48 |
| $FGSM_2$ | 1.41 | 0.6 | 0.51 | 1.39 | 0.9 | 0.16 | 0.5 | 1.42 |
| $PGD_1$ | 14.15 | 7.76 | 5.83 | 15.3 | 3.42 | 1.6 | 3.03 | 13.94 |
| $PGD_2$ | 0.02 | 0 | 0 | 0 | 0.19 | 0.09 | 0 | 0.02 |
| $PGD_3$ | 14.3 | 8.05 | 5.83 | 15.5 | 3.52 | 1.67 | 3.11 | 14.33 |
| $PGD_4$ | 0.01 | 0 | 0 | 0 | 0.17 | 0.07 | 0 | 0.01 |
| $PGD_5$ | 0 | 0.01 | 0 | 0 | 0.19 | 0 | 0.02 | 0 |
| DIFGSM | 33.68 | 24.49 | 28.83 | 32.76 | 17.55 | 14.05 | 22.04 | 30.92 |
| CW | 0 | 0 | 0 | 0 | 0 | 0 | 0 | 0 |
| Jitter | 44.21 | 36.41 | 45.44 | 42.21 | 26.4 | 23.85 | 40.67 | 43.86 |
| TIFGSM | 44.26 | 35.39 | 39.8 | 43.84 | 27.62 | 22.01 | 34.77 | 44 |
| PIFGSM | 6.32 | 3.3 | 1.54 | 8.54 | 0.75 | 0.6 | 0.65 | 6.39 |
| EADEN | 0 | 0 | 0 | 0 | 0 | 0 | 0 | 0 |
| OnePixel | 60.73 | 53.74 | 61.77 | 59.99 | 39.56 | 39.56 | 55.33 | 60.63 |
| Pixle | 6.63 | 5.1 | 5.02 | 8.17 | 4.14 | 1.92 | 2.3 | 6.33 |
| SPSA | 54.22 | 45.44 | 51.11 | 54.88 | 30.33 | 31.59 | 44.47 | 54.05 |
| Square | 0.06 | 0.01 | 0 | 0.04 | 0 | 0 | 0.01 | 0.05 |
| TAP | 63.71 | 56.62 | 63.75 | 63.26 | 42.74 | 32.92 | 58.79 | 63.51 |
| *ASV* | 26.99 | 36.64 | 22.96 | 33.93 | 30.43 | 9.08 | 24.15 | 26.51 |
| *FFF (no-data)* | 27.53 | 36.41 | 22.11 | 32.81 | 16.84 | 9.27 | 24.65 | 26.85 |
| *FFF (mean-std)* | 27.43 | 36.96 | 23.31 | 32.79 | 20.61 | 9.54 | 25.06 | 26.87 |
| *FFF (one-sample)* | 27.45 | 36.67 | 22.75 | 33.16 | 18.17 | 9.23 | 24.71 | 26.89 |
| *FG-UAP* | 24.44 | 35.57 | 17.41 | 29.54 | 3.31 | 7.83 | 20.54 | 24.03 |
| *GD-UAP (no-data)* | 27.12 | 36.82 | 22.16 | 32.86 | 22.32 | 9.28 | 24.71 | 27.00 |
| *GD-UAP (mean-std)* | 27.30 | 37.18 | 22.62 | 33.14 | 22.30 | 9.69 | 25.36 | 26.97 |
| *GD-UAP (one-sample)* | 27.61 | 36.84 | 22.42 | 33.09 | 22.11 | 9.24 | 24.91 | 26.87 |
| *L4A-base* | 26.40 | 36.71 | 22.98 | 32.66 | 5.17 | 9.43 | 22.75 | 26.09 |
| *L4A-fuse* | 26.65 | 36.69 | 22.82 | 32.50 | 5.35 | 9.55 | 22.75 | 26.30 |
| *L4A-ugs* | 27.42 | 37.33 | 23.49 | 33.19 | 29.03 | 9.43 | 25.67 | 27.14 |
| *PD-UAP* | 27.21 | 37.12 | 22.83 | 33.09 | 23.87 | 9.39 | 25.51 | 26.92 |
| *SSP* | 25.18 | 36.15 | 22.45 | 31.94 | 16.58 | 8.85 | 24.03 | 25.08 |
| *STD* | 26.81 | 36.34 | 21.81 | 33.27 | 14.05 | 9.55 | 24.85 | 26.58 |
| *UAP (DeepFool)* | 27.52 | 37.07 | 23.60 | 33.44 | 12.91 | 9.56 | 25.56 | 27.01 |
| *UAPEPGD* | 27.32 | 37.01 | 23.54 | 33.75 | 26.12 | 9.50 | 25.69 | 27.21 |
| Clean Accuracy | 64.2 | 57.62 | 65.62 | 63.61 | 43.81 | 47.1 | 59.78 | 64.12 |
| IAA Avg. | 19.90 ↓69% | 15.84 ↓73% | 17.54 ↓73% | 20.12 ↓68% | 11.14 ↓75% | 9.56 ↓80% | 14.95 ↓75% | 19.66 ↓69% |
| UAP Avg. | 26.89 ↓58% | 36.71 ↓36% | 22.45 ↓66% | 32.82 ↓48% | 18.07 ↓59% | 9.27 ↓80% | 24.43 ↓59% | 26.52 ↓59% |
| Adv Avg. | 23.19 ↓64% | 25.66 ↓55% | 19.85 ↓70% | 26.09 ↓60% | 14.40 ↓67% | 9.42 ↓80% | 19.41 ↓68% | 22.89 ↓64% |

### B.7.4  CIFAR 10

Table 16: This table presents the results of various instance and universal adversarial perturbation (UAP) attacks on the CIFAR 10 dataset, with all UAP attack names in *italics*. Different configurations of FGSM and PGD are denoted, such as $FGSM_1$ and $PGD_1$. Average results for universal adversarial perturbations (UAP Avg.), instance adversarial attacks (IAA Avg.), and overall adversarial performance (Adv Avg.) are reported at the bottom, including percentage drops relative to clean accuracy.

| | Barlow | BYOL | DINO | MoCoV3 | SimCLR | Supervised | SwAV | VICReg |
|---|---|---|---|---|---|---|---|---|
| $FGSM_1$ | 32.95 | 31.04 | 27.57 | 33.04 | 37.86 | 42.84 | 19.38 | 33.04 |
| $FGSM_2$ | 53.83 | 50.24 | 52.58 | 52.51 | 59.88 | 29.71 | 47.54 | 53.94 |
| $PGD_1$ | 34.76 | 29.2 | 22.25 | 35.16 | 23.51 | 36.92 | 21.04 | 34.64 |
| $PGD_2$ | 0.02 | 0 | 0 | 0 | 0.03 | 0 | 0 | 0.01 |
| $PGD_3$ | 34.02 | 28.38 | 20.85 | 34.44 | 22.48 | 36.51 | 20.71 | 34.23 |
| $PGD_4$ | 0.02 | 0.02 | 0 | 0 | 0.03 | 0 | 0 | 0 |
| $PGD_5$ | 0 | 0 | 0 | 0 | 0.01 | 0 | 0 | 0 |
| DIFGSM | 56.24 | 52.78 | 42.53 | 55.48 | 52.39 | 55.9 | 39.2 | 54.64 |
| CW | 0 | 0 | 0 | 0 | 0.06 | 0 | 0 | 0 |
| Jitter | 66.67 | 62.37 | 59.8 | 66.97 | 55.15 | 70.7 | 58.5 | 67.63 |
| TIFGSM | 52.32 | 48.88 | 41.23 | 50.64 | 56.11 | 56.88 | 42.38 | 54.51 |
| PIFGSM | 0.39 | 0.22 | 0.04 | 0.28 | 0.45 | 5.18 | 0 | 0.41 |
| EADEN | 0 | 0 | 0 | 0 | 0 | 0 | 0 | 0 |
| OnePixel | 87.36 | 86.09 | 88.42 | 87.78 | 82.28 | 85.59 | 81.11 | 87.21 |
| Pixle | 5.55 | 2.15 | 4.44 | 3.02 | 1.82 | 2.22 | 1.93 | 5.41 |
| SPSA | 69.6 | 79.09 | 55.69 | 80.73 | 60.51 | 71.34 | 68.81 | 69.9 |
| Square | 0 | 0 | 0.05 | 0 | 0 | 0 | 0 | 0 |
| TAP | 88.51 | 91.06 | 89.82 | 91.4 | 87.56 | 77.66 | 92.14 | 88.59 |
| *ASV* | 43.79 | 57.79 | 38.25 | 51.44 | 49.14 | 33.42 | 44.43 | 44.01 |
| *FFF (no-data)* | 44.64 | 55.64 | 31.33 | 42.83 | 19.94 | 31.94 | 41.00 | 45.20 |
| *FFF (mean-std)* | 47.57 | 49.83 | 31.22 | 43.39 | 10.22 | 27.04 | 31.44 | 47.28 |
| *FFF (one-sample)* | 47.08 | 55.60 | 33.14 | 45.44 | 10.41 | 28.45 | 36.48 | 47.15 |
| *FG-UAP* | 25.94 | 45.95 | 13.50 | 12.52 | 10.19 | 16.20 | 11.27 | 25.72 |
| *GD-UAP (no-data)* | 44.25 | 50.00 | 32.89 | 44.02 | 19.16 | 33.33 | 39.96 | 44.85 |
| *GD-UAP (mean-std)* | 45.92 | 53.10 | 30.33 | 35.97 | 10.15 | 27.96 | 27.24 | 44.08 |
| *GD-UAP (one-sample)* | 47.37 | 56.57 | 33.32 | 47.55 | 14.90 | 29.48 | 39.60 | 47.62 |
| *L4A-base* | 44.50 | 40.64 | 37.60 | 40.09 | 10.46 | 27.73 | 17.49 | 45.44 |
| *L4A-fuse* | 45.01 | 41.02 | 38.03 | 40.84 | 10.31 | 27.71 | 17.03 | 44.95 |
| *L4A-ugs* | 48.25 | 60.94 | 36.47 | 47.30 | 56.77 | 31.15 | 41.07 | 48.86 |
| *PD-UAP* | 48.29 | 58.88 | 31.88 | 51.28 | 49.43 | 29.52 | 39.50 | 49.65 |
| *SSP* | 24.65 | 27.38 | 34.23 | 17.89 | 12.86 | 22.63 | 28.44 | 25.18 |
| *STD* | 45.28 | 59.12 | 27.10 | 45.58 | 52.44 | 34.53 | 40.51 | 45.37 |
| *UAP (DeepFool)* | 48.22 | 57.93 | 37.35 | 51.53 | 10.34 | 33.31 | 40.21 | 48.83 |
| *UAPEPGD* | 48.16 | 57.77 | 37.89 | 53.14 | 57.77 | 34.05 | 45.79 | 48.52 |
| Clean Accuracy | 92.78 | 93.05 | 93.85 | 94.67 | 90.98 | 91.4 | 93.9 | 92.79 |
| IAA Avg. | 32.34 ↓65% | 31.19↓66% | 28.07↓70% | 32.85↓65% | 30.00↓67% | 31.74↓65% | 27.37↓71% | 32.45↓65% |
| UAP Avg. | 43.68 ↓53% | 51.76↓44% | 32.78↓65% | 41.92↓56% | 25.28↓72% | 29.27↓68% | 33.84↓64% | 43.91↓53% |
| Adv Avg. | 37.68 ↓59% | 40.87↓56% | 30.28↓68% | 37.12↓61% | 27.78↓69% | 30.58↓66% | 30.41↓68% | 37.84↓59% |

### B.7.5 CIFAR 100

Table 17: This table presents the results of various instance and universal adversarial perturbation (UAP) attacks on the CIFAR 100 dataset, with all UAP attack names in *italics*. Different configurations of FGSM and PGD are denoted, such as $FGSM_1$ and $PGD_1$. Average results for universal adversarial perturbations (UAP Avg.), instance adversarial attacks (IAA Avg.), and overall adversarial performance (Adv Avg.) are reported at the bottom, including percentage drops relative to clean accuracy.

| | Barlow | BYOL | DINO | MoCoV3 | SimCLR | Supervised | SwAV | VICReg |
|---|---|---|---|---|---|---|---|---|
| $FGSM_1$ | 20.52 | 19.01 | 16.03 | 19.29 | 19.49 | 24.51 | 11.07 | 22.34 |
| $FGSM_2$ (e=1) | 34.07 | 31.02 | 34.08 | 28.84 | 30.06 | 18.20 | 29.16 | 35.71 |
| $PGD_1$ | 19.74 | 14.42 | 11.47 | 18.38 | 8.92 | 19.09 | 10.29 | 20.98 |
| $PGD_2$ | 0.04 | 0 | 0 | 0.02 | 0.12 | 0 | 0.01 | 0.06 |
| $PGD_3$) | 19.33 | 14.18 | 11.09 | 17.69 | 8.24 | 18.85 | 9.92 | 20.67 |
| $PGD_4$ | 0.06 | 0.01 | 0 | 0.01 | 0.08 | 0 | 0 | 0.02 |
| $PGD_5$ | 0 | 0 | 0 | 0 | 0.18 | 0.01 | 0 | 0 |
| DIFGSM | 38.20 | 35.26 | 27.54 | 32.23 | 32.56 | 34.97 | 26.31 | 39.47 |
| CW | 0.01 | 0 | 0 | 0.06 | 0.02 | 0.02 | 0 | 0.04 |
| Jitter | 66.85 | 62.15 | 59.33 | 65.89 | 42.01 | 67.10 | 53.82 | 66.73 |
| TIFGSM | 34.84 | 36.35 | 27.80 | 30.79 | 35.15 | 36.82 | 29.15 | 37.30 |
| PIFGSM | 0.78 | 0.34 | 0.17 | 0.58 | 0.36 | 3.29 | 0.09 | 1.10 |
| EADEN | 0 | 0 | 0 | 0 | 0 | 0 | 0 | 0 |
| OnePixel | 67.73 | 66.25 | 69.87 | 67.64 | 58.73 | 64.76 | 61.41 | 68.19 |
| Pixle | 0.48 | 0.96 | 0.56 | 0.90 | 0.96 | 1.40 | 0.43 | 0.55 |
| SPSA | 47.25 | 54.96 | 38.62 | 53.70 | 28.82 | 48.30 | 44.86 | 49.46 |
| Square | 0.06 | 0.01 | 0.04 | 0 | 0 | 0 | 0.01 | 0.05 |
| TAP | 70.29 | 72.88 | 69.76 | 73.97 | 65.1 | 53.57 | 76.14 | 70.29 |
| *ASV* | 24.05 | 37.80 | 19.72 | 26.52 | 24.66 | 16.70 | 27.82 | 25.10 |
| *FFF (no-data)* | 25.38 | 37.78 | 16.33 | 21.70 | 9.33 | 15.49 | 26.25 | 26.75 |
| *FFF (mean-std)* | 27.59 | 35.74 | 13.40 | 23.23 | 2.01 | 12.73 | 20.63 | 27.45 |
| *FFF (one-sample)* | 26.64 | 37.25 | 17.63 | 22.56 | 3.19 | 14.06 | 24.63 | 27.48 |
| *FG-UAP* | 12.51 | 29.98 | 3.67 | 9.89 | 1.17 | 8.55 | 3.61 | 12.85 |
| *GD-UAP (no-data)* | 25.30 | 36.37 | 16.82 | 21.75 | 10.45 | 16.15 | 24.53 | 26.23 |
| *GD-UAP (mean-std)* | 26.15 | 36.98 | 15.39 | 20.84 | 3.64 | 13.78 | 19.36 | 27.09 |
| *GD-UAP (one-sample)* | 26.82 | 37.82 | 17.85 | 23.42 | 4.32 | 14.58 | 25.77 | 27.56 |
| *L4A-base* | 27.10 | 28.94 | 18.05 | 21.41 | 1.10 | 14.72 | 8.38 | 28.24 |
| *L4A-fuse* | 27.50 | 28.92 | 18.22 | 21.72 | 1.25 | 14.67 | 8.51 | 27.67 |
| *L4A-ugs* | 28.78 | 39.53 | 19.65 | 24.87 | 28.25 | 15.66 | 26.49 | 29.24 |
| *PD-UAP* | 27.85 | 39.75 | 15.92 | 25.05 | 22.89 | 14.44 | 26.49 | 28.91 |
| *SSP* | 13.18 | 21.37 | 19.00 | 12.58 | 6.38 | 10.64 | 16.14 | 13.64 |
| *STD* | 25.27 | 38.29 | 13.29 | 21.74 | 15.43 | 16.96 | 25.74 | 26.31 |
| *UAP (DeepFool)* | 27.04 | 38.31 | 19.64 | 25.68 | 2.94 | 16.43 | 26.17 | 28.41 |
| *UAPEPGD* | 26.65 | 37.62 | 19.84 | 27.39 | 28.27 | 17.44 | 28.42 | 28.29 |
| Clean Accuracy | 77.86 | 78.18 | 76.67 | 80.19 | 72.97 | 73.86 | 79.41 | 77.79 |
| IAA Avg. | 23.34 ↓70% | 22.65↓71% | 20.45↓74% | 22.77↓72% | 18.36↓75% | 21.72↓71% | 19.59↓75% | 24.05↓69% |
| UAP Avg. | 24.86 ↓68% | 35.15↓55% | 16.52↓79% | 21.89↓73% | 10.33↓86% | 14.56↓80% | 21.18↓73% | 25.70↓67% |
| Adv Avg. | 24.06 ↓69% | 28.53↓63% | 18.55↓77% | 22.36↓72% | 14.58↓80% | 18.34↓75% | 20.34↓74% | 24.82↓68% |

### B.7.6   DTD

Table 18: This table presents the results of various instance and universal adversarial perturbation (UAP) attacks on the DTD dataset, with all UAP attack names in *italics*. Different configurations of FGSM and PGD are denoted, such as $FGSM_1$ and $PGD_1$. Average results for universal adversarial perturbations (UAP Avg.), instance adversarial attacks (IAA Avg.), and overall adversarial performance (Adv Avg.) are reported at the bottom, including percentage drops relative to clean accuracy.

| | Barlow | BYOL | DINO | MoCoV3 | SimCLR | Supervised | SwAV | VICReg |
|---|---|---|---|---|---|---|---|---|
| $FGSM_1$ | 50.43 | 46.76 | 48.88 | 51.65 | 38.4 | 42.02 | 48.99 | 52.71 |
| $FGSM_2$ | 23.24 | 21.28 | 23.94 | 24.63 | 17.87 | 17.66 | 25.80 | 26.54 |
| $PGD_1$ | 50.05 | 46.01 | 47.93 | 51.17 | 39.31 | 40.05 | 48.99 | 51.65 |
| $PGD_2$ | 6.91 | 4.57 | 3.46 | 6.38 | 2.13 | 3.19 | 3.35 | 6.91 |
| $PGD_3$ | 50.11 | 46.54 | 48.14 | 51.17 | 39.04 | 40.27 | 48.62 | 51.65 |
| $PGD_4$ | 6.54 | 3.94 | 2.82 | 5.96 | 1.7 | 2.93 | 3.03 | 6.60 |
| $PGD_5$ | 14.89 | 12.23 | 11.81 | 16.76 | 3.99 | 10.37 | 10.53 | 16.22 |
| DIFGSM | 59.84 | 52.87 | 60.05 | 59.79 | 52.02 | 54.47 | 60.27 | 64.20 |
| CW | 0.32 | 0.32 | 0.74 | 0.69 | 0.43 | 0.64 | 0.90 | 0.90 |
| Jitter | 67.39 | 65.90 | 66.91 | 66.17 | 62.02 | 60.48 | 68.51 | 68.30 |
| TIFGSM | 67.77 | 65.32 | 67.93 | 66.06 | 62.07 | 62.34 | 67.39 | 68.88 |
| PIFGSM | 42.77 | 38.83 | 40.16 | 45.53 | 26.76 | 35.43 | 38.40 | 43.94 |
| EADEN | 0 | 0 | 0 | 0 | 0 | 0 | 0 | 0 |
| OnePixel | 75.32 | 75.43 | 76.17 | 74.41 | 71.12 | 70.69 | 75.96 | 76.28 |
| Pixle | 49.89 | 46.28 | 46.97 | 49.57 | 40.48 | 37.62 | 41.38 | 50.90 |
| SPSA | 72.87 | 71.81 | 73.51 | 72.39 | 67.98 | 66.91 | 73.78 | 74.15 |
| Square | 8.09 | 5.96 | 6.7 | 7.77 | 5.74 | 1.49 | 8.46 | 8.67 |
| TAP | 74.10 | 73.78 | 73.72 | 72.50 | 72.07 | 62.98 | 76.97 | 75.05 |
| *ASV* | 53.19 | 61.97 | 49.31 | 56.54 | 67.39 | 39.04 | 58.19 | 54.04 |
| *FFF (no-data)* | 53.24 | 61.60 | 48.24 | 56.33 | 54.89 | 38.24 | 57.13 | 54.31 |
| *FFF (mean-std)* | 52.55 | 61.33 | 48.51 | 56.01 | 56.54 | 38.35 | 57.98 | 53.72 |
| *FFF (one-sample)* | 52.87 | 61.60 | 49.26 | 56.76 | 52.18 | 38.35 | 57.55 | 54.15 |
| *FG-UAP* | 52.77 | 61.17 | 46.12 | 55.43 | 21.38 | 36.76 | 53.62 | 53.67 |
| *GD-UAP (no-data)* | 53.24 | 61.86 | 48.78 | 56.38 | 55.53 | 38.30 | 57.93 | 54.10 |
| *GD-UAP (mean-std)* | 52.77 | 60.96 | 48.88 | 55.59 | 62.18 | 38.72 | 57.29 | 53.99 |
| *GD-UAP (one-sample)* | 52.87 | 62.34 | 49.31 | 56.38 | 55.80 | 38.40 | 57.87 | 54.04 |
| *L4A-base* | 51.86 | 61.44 | 49.63 | 57.34 | 29.52 | 38.83 | 55.80 | 52.71 |
| *L4A-fuse* | 51.91 | 61.54 | 49.04 | 56.76 | 30.32 | 38.46 | 55.32 | 52.66 |
| *L4A-ugs* | 52.71 | 61.76 | 49.47 | 56.86 | 59.04 | 38.83 | 57.87 | 53.30 |
| *PD-UAP* | 53.40 | 61.65 | 48.94 | 57.29 | 65.64 | 38.78 | 57.98 | 54.31 |
| *SSP* | 52.23 | 61.60 | 48.46 | 55.59 | 53.35 | 36.81 | 57.02 | 52.93 |
| *STD* | 53.14 | 61.97 | 49.10 | 56.12 | 60.27 | 39.04 | 58.35 | 54.15 |
| *UAP (DeepFool)* | 53.51 | 61.86 | 49.47 | 56.97 | 54.04 | 38.99 | 58.14 | 54.47 |
| *UAPEPGD* | 53.40 | 61.76 | 49.63 | 56.81 | 69.31 | 39.26 | 58.14 | 53.94 |
| Clean Accuracy | 79.97 | 76.76 | 77.02 | 75.43 | 73.19 | 72.13 | 77.45 | 77.61 |
| IAA Avg. | 40.02 ↓50% | 37.65↓51% | 38.88↓50% | 40.14↓50% | 33.50↓54% | 33.86↓53% | 38.96↓50% | 41.30↓47% |
| UAP Avg. | 52.85 ↓34% | 61.65↓17% | 48.88↓37% | 56.44↓25% | 52.96↓28% | 38.44↓47% | 57.26↓26% | 53.78↓31% |
| Adv Avg. | 46.06 ↓42% | 48.94↓34% | 43.58↓43% | 47.81↓37% | 42.66↓42% | 36.02↓50% | 47.57↓39% | 47.17↓39% |

### B.7.7 FLOWERS

Table 19: This table presents the results of various instance and universal adversarial perturbation (UAP) attacks on the Flowers dataset, with all UAP attack names in *italics*. Different configurations of FGSM and PGD are denoted, such as $FGSM_1$ and $PGD_1$. Average results for universal adversarial perturbations (UAP Avg.), instance adversarial attacks (IAA Avg.), and overall adversarial performance (Adv Avg.) are reported at the bottom, including percentage drops relative to clean accuracy.

| | Barlow | BYOL | DINO | MocoV3 | SimCLR | Supervised | SwAV | VICReg |
|---|---|---|---|---|---|---|---|---|
| $FGSM_1$ | 66.36 | 57.69 | 57.37 | 64.52 | 48.50 | 41.85 | 46.97 | 66.36 |
| $FGSM_2$ | 25.96 | 17.49 | 19.44 | 24.96 | 19.00 | 7.68 | 13.33 | 25.96 |
| $PGD_1$ | 66.03 | 55.99 | 55.60 | 63.31 | 50.45 | 36.97 | 46.65 | 65.81 |
| $PGD_2$ | 1.51 | 0.37 | 0.17 | 1.10 | 0.15 | 0.00 | 0.06 | 1.65 |
| $PGD_3$ | 66.19 | 56.37 | 55.95 | 63.50 | 51.00 | 37.31 | 46.72 | 66.44 |
| $PGD_4$ | 1.21 | 0.38 | 0.13 | 0.90 | 0.13 | 0.00 | 0.02 | 1.29 |
| $PGD_5$ | 8.03 | 4.90 | 2.81 | 7.17 | 0.92 | 0.72 | 0.89 | 8.05 |
| DI2FGSM | 74.42 | 72.08 | 69.73 | 75.75 | 62.56 | 56.94 | 67.56 | 78.12 |
| CW | 0.00 | 0.00 | 0.05 | 0.00 | 0.00 | 0.02 | 0.00 | 0.00 |
| Jitter | 84.93 | 80.12 | 81.87 | 82.53 | 79.85 | 73.62 | 79.24 | 84.33 |
| TIFGSM | 86.85 | 84.35 | 87.48 | 86.17 | 81.29 | 75.36 | 84.39 | 87.88 |
| PIFGSM | 53.81 | 43.06 | 39.04 | 51.65 | 29.16 | 27.46 | 28.63 | 53.85 |
| EADEN | 0.00 | 0.00 | 0.00 | 0.00 | 0.00 | 0.00 | 0.00 | 0.00 |
| OnePixel | 94.47 | 92.77 | 94.79 | 93.10 | 89.27 | 88.38 | 92.94 | 94.49 |
| Pixle | 35.32 | 38.34 | 31.21 | 45.08 | 32.07 | 20.88 | 24.05 | 35.09 |
| SPSA | 93.03 | 90.21 | 92.91 | 91.84 | 85.56 | 84.31 | 90.60 | 92.84 |
| Square | 6.70 | 4.17 | 4.40 | 5.60 | 4.90 | 0.06 | 3.32 | 6.70 |
| TAP | 94.01 | 92.77 | 94.76 | 93.31 | 89.76 | 75.93 | 93.14 | 94.01 |
| *ASV* | 74.65 | 81.96 | 69.86 | 75.20 | 76.95 | 34.19 | 71.70 | 74.54 |
| *FFF (no-data)* | 75.06 | 81.66 | 67.49 | 75.34 | 54.67 | 33.26 | 70.17 | 74.73 |
| *FFF (mean-std)* | 74.37 | 82.00 | 68.08 | 75.18 | 56.34 | 34.44 | 72.49 | 74.89 |
| *FFF (one-sample)* | 74.27 | 81.51 | 68.22 | 75.15 | 57.85 | 33.41 | 71.45 | 74.32 |
| *FG-UAP* | 72.29 | 80.97 | 59.92 | 71.93 | 24.88 | 29.85 | 55.78 | 72.22 |
| *GD-UAP (no-data)* | 74.57 | 81.97 | 67.77 | 75.37 | 60.37 | 33.90 | 71.27 | 74.95 |
| *GD-UAP (mean-std)* | 74.66 | 81.97 | 68.44 | 75.31 | 69.39 | 34.35 | 71.95 | 75.26 |
| *GD-UAP (one-sample)* | 74.47 | 81.39 | 67.98 | 75.21 | 61.25 | 33.92 | 71.56 | 74.18 |
| *L4A-base* | 73.46 | 81.92 | 69.25 | 75.01 | 25.70 | 34.75 | 67.80 | 73.14 |
| *L4A-fuse* | 73.43 | 81.98 | 69.16 | 75.27 | 25.91 | 34.73 | 67.23 | 73.33 |
| *L4A-ugs* | 74.81 | 81.95 | 70.51 | 75.66 | 76.75 | 34.42 | 72.67 | 74.47 |
| *PD-UAP* | 74.17 | 82.46 | 68.98 | 75.16 | 74.75 | 34.08 | 71.68 | 73.68 |
| *SSP* | 73.07 | 81.27 | 65.60 | 73.38 | 56.14 | 32.28 | 66.85 | 73.28 |
| *STD* | 73.81 | 81.54 | 66.81 | 74.11 | 51.96 | 33.64 | 71.12 | 73.56 |
| *UAP (DeepFool)* | 75.15 | 82.41 | 70.32 | 75.98 | 41.61 | 34.81 | 73.18 | 75.02 |
| *UAPEPGD* | 75.70 | 82.54 | 70.47 | 76.28 | 81.67 | 35.32 | 73.33 | 75.76 |
| Clean Accuracy | 94.92 | 93.36 | 95.23 | 94.07 | 90.57 | 90.59 | 93.84 | 94.92 |
| IAA Avg. | 47.71 ↓50% | 43.94 ↓53% | 43.76 ↓54% | 47.25 ↓50% | 40.25 ↓56% | 34.86 ↓62% | 39.92 ↓58% | 47.94 ↓50% |
| UAP Avg. | 74.25 ↓22% | 81.84 ↓12% | 68.05 ↓29% | 74.97 ↓20% | 56.01 ↓38% | 33.83 ↓63% | 70.01 ↓25% | 74.20 ↓22% |
| Adv Avg. | 60.19 ↓37% | 61.78 ↓34% | 55.19 ↓42% | 60.30 ↓36% | 47.66 ↓47% | 34.37 ↓62% | 54.08 ↓42% | 60.30 ↓37% |

### B.7.8  FOOD

Table 20: This table presents the results of various instance and universal adversarial perturbation (UAP) attacks on the Food dataset, with all UAP attack names in *italics*. Different configurations of FGSM and PGD are denoted, such as $FGSM_1$ and $PGD_1$. Average results for universal adversarial perturbations (UAP Avg.), instance adversarial attacks (IAA Avg.), and overall adversarial performance (Adv Avg.) are reported at the bottom, including percentage drops relative to clean accuracy.

|  | Barlow | BYOL | DINO | MocoV3 | SimCLR | Supervised | SwAV | VICReg |
|---|---|---|---|---|---|---|---|---|
| $FGSM_1$ | 26.40 | 19.34 | 14.13 | 28.69 | 12.10 | 13.18 | 12.95 | 23.48 |
| $FGSM_2$ | 3.24 | 1.50 | 1.39 | 4.02 | 1.41 | 1.29 | 0.95 | 2.52 |
| $PGD_1$ | 26.60 | 19.03 | 13.87 | 28.54 | 13.69 | 11.30 | 13.15 | 23.91 |
| $PGD_2$ | 0.04 | 0.01 | 0.01 | 0.05 | 0.00 | 0.02 | 0.00 | 0.04 |
| $PGD_3$ | 26.72 | 19.21 | 14.13 | 28.76 | 13.92 | 11.42 | 13.48 | 24.12 |
| $PGD_4$ | 0.04 | 0.01 | 0.00 | 0.04 | 0.00 | 0.01 | 0.00 | 0.03 |
| $PGD_5$ | 0.59 | 0.19 | 0.10 | 0.82 | 0.04 | 0.13 | 0.01 | 0.47 |
| DI2FGSM | 44.15 | 37.23 | 37.35 | 44.94 | 33.02 | 32.45 | 37.32 | 40.14 |
| CW | 0.00 | 0.00 | 0.00 | 0.00 | 0.00 | 0.00 | 0.00 | 0.00 |
| Jitter | 60.70 | 56.00 | 61.34 | 58.14 | 55.13 | 53.14 | 61.79 | 59.79 |
| TIFGSM | 57.43 | 51.93 | 53.38 | 56.41 | 48.65 | 45.76 | 54.04 | 56.51 |
| PIFGSM | 17.53 | 11.71 | 6.67 | 19.93 | 5.17 | 6.80 | 5.46 | 14.85 |
| EADEN | 0.00 | 0.00 | 0.00 | 0.00 | 0.00 | 0.00 | 0.00 | 0.00 |
| OnePixel | 73.54 | 69.95 | 76.00 | 71.41 | 63.59 | 64.63 | 73.65 | 73.22 |
| Pixle | 14.94 | 12.97 | 9.65 | 17.11 | 8.34 | 5.84 | 4.93 | 13.66 |
| SPSA | 68.75 | 64.12 | 69.31 | 66.70 | 57.49 | 56.88 | 67.11 | 68.04 |
| Square | 0.19 | 0.05 | 0.09 | 0.19 | 0.16 | 0.02 | 0.07 | 0.16 |
| TAP | 74.21 | 71.43 | 76.25 | 72.68 | 65.96 | 53.74 | 76.18 | 73.75 |
| *ASV* | 40.26 | 49.36 | 39.71 | 43.88 | 51.44 | 19.71 | 45.23 | 39.17 |
| *FFF (no-data)* | 40.46 | 48.56 | 38.05 | 43.07 | 33.07 | 19.15 | 43.68 | 39.47 |
| *FFF (mean-std)* | 40.22 | 49.06 | 38.14 | 43.01 | 35.53 | 19.43 | 44.90 | 39.28 |
| *FFF (one-sample)* | 40.25 | 49.00 | 38.50 | 43.29 | 31.72 | 19.37 | 44.82 | 39.22 |
| *FG-UAP* | 38.01 | 46.65 | 34.36 | 39.87 | 3.53 | 16.70 | 35.61 | 36.82 |
| *GD-UAP (no-sample)* | 40.53 | 49.04 | 38.40 | 43.19 | 36.96 | 19.35 | 44.31 | 39.42 |
| *GD-UAP (mean-std)* | 40.10 | 48.83 | 38.04 | 43.08 | 45.96 | 19.62 | 44.95 | 39.40 |
| *GD-UAP (one-sample)* | 40.30 | 48.78 | 38.70 | 43.36 | 35.96 | 19.42 | 44.88 | 39.33 |
| *L4A-base* | 39.47 | 48.97 | 39.04 | 43.27 | 5.96 | 19.84 | 41.26 | 38.50 |
| *L4A-fuse* | 39.47 | 48.90 | 39.01 | 43.24 | 6.12 | 19.98 | 41.26 | 38.39 |
| *L4A-ugs* | 40.65 | 49.21 | 39.64 | 43.73 | 45.19 | 19.82 | 45.33 | 39.47 |
| *PD-UAP* | 39.94 | 49.32 | 38.86 | 43.34 | 50.08 | 19.59 | 44.59 | 38.98 |
| *SSP* | 39.30 | 47.60 | 37.02 | 41.92 | 30.96 | 17.84 | 42.59 | 38.31 |
| *STD* | 39.87 | 48.60 | 37.86 | 42.91 | 36.31 | 19.48 | 44.28 | 38.88 |
| *UAP (DeepFool)* | 40.87 | 49.29 | 39.57 | 44.12 | 21.17 | 20.23 | 45.69 | 39.81 |
| *UAPEPGD* | 41.07 | 49.78 | 39.81 | 44.27 | 57.10 | 20.24 | 46.32 | 40.11 |
| Clean Accuracy | 76.09 | 73.07 | 78.42 | 73.83 | 67.24 | 69.05 | 76.51 | 75.81 |
| IAA Avg. | 27.50 ↓64% | 24.15↓67% | 24.09↓69% | 27.69↓62% | 21.03↓69% | 19.81↓71% | 23.39↓69% | 26.37↓65% |
| UAP Avg. | 40.04 ↓47% | 48.81↓33% | 38.41↓51% | 43.09↓42% | 32.94↓51% | 19.36↓72% | 43.73↓43% | 39.03↓49% |
| Adv Avg. | 33.40 ↓56% | 35.75↓51% | 30.83↓61% | 34.94↓53% | 26.63↓60% | 19.59↓72% | 32.96↓57% | 32.33↓57% |

### B.7.9 PETS

Table 21: This table presents the results of various instance and universal adversarial perturbation (UAP) attacks on the Pets dataset, with all UAP attack names in *italics*. Different configurations of FGSM and PGD are denoted, such as $FGSM_1$ and $PGD_1$. Average results for universal adversarial perturbations (UAP Avg.), instance adversarial attacks (IAA Avg.), and overall adversarial performance (Adv Avg.) are reported at the bottom, including percentage drops relative to clean accuracy.

| Method | Barlow | BYOL | DINO | MocoV3 | SimCLR | Supervised | SwAV | VICReg |
|---|---|---|---|---|---|---|---|---|
| $FGSM_1$ | 63.58 | 61.00 | 48.74 | 71.38 | 44.60 | 55.10 | 41.59 | 63.58 |
| $FGSM_2$ | 25.08 | 21.62 | 11.81 | 34.65 | 17.20 | 14.17 | 8.74 | 25.08 |
| $PGD_1$ | 64.38 | 60.82 | 48.07 | 71.07 | 46.76 | 52.20 | 43.00 | 64.30 |
| $PGD_2$ | 0.82 | 0.41 | 0.08 | 2.96 | 0.16 | 0.00 | 0.03 | 0.79 |
| $PGD_3$ | 64.52 | 61.21 | 48.10 | 71.29 | 47.25 | 52.21 | 43.42 | 64.52 |
| $PGD_4$ | 0.63 | 0.27 | 0.03 | 2.39 | 0.11 | 0.00 | 0.03 | 0.57 |
| $PGD_5$ | 6.54 | 5.69 | 0.89 | 14.03 | 0.98 | 1.38 | 0.43 | 6.51 |
| DI2FGSM | 73.92 | 71.18 | 63.92 | 78.63 | 61.06 | 68.25 | 59.70 | 74.18 |
| CW | 0.00 | 0.00 | 0.00 | 0.03 | 0.00 | 0.00 | 0.00 | 0.00 |
| Jitter | 80.75 | 79.82 | 75.82 | 84.06 | 74.50 | 78.41 | 75.60 | 80.83 |
| TIFGSM | 81.43 | 80.30 | 78.13 | 84.89 | 75.31 | 80.60 | 76.11 | 82.31 |
| PIFGSM | 54.24 | 51.35 | 34.23 | 64.67 | 31.70 | 41.02 | 26.11 | 54.24 |
| EADEN | 0.00 | 0.00 | 0.00 | 0.00 | 0.00 | 0.00 | 0.00 | 0.00 |
| OnePixel | 88.28 | 87.90 | 87.65 | 89.82 | 81.51 | 90.60 | 85.85 | 88.31 |
| Pixle | 42.04 | 41.46 | 38.47 | 59.12 | 31.61 | 43.01 | 29.16 | 42.31 |
| SPSA | 87.27 | 86.87 | 85.81 | 88.79 | 79.93 | 88.54 | 83.95 | 87.46 |
| Square | 3.28 | 1.71 | 0.49 | 4.79 | 3.88 | 0.05 | 0.46 | 3.30 |
| TAP | 88.97 | 89.04 | 88.41 | 90.66 | 83.01 | 86.83 | 87.09 | 88.97 |
| *ASV* | 62.97 | 75.13 | 62.88 | 70.10 | 78.67 | 49.80 | 66.06 | 63.20 |
| *FFF (no-data)* | 63.42 | 75.13 | 62.52 | 69.77 | 64.93 | 48.89 | 65.52 | 63.43 |
| *FFF (mean-std)* | 63.24 | 75.17 | 62.07 | 70.03 | 68.70 | 49.81 | 66.13 | 63.17 |
| *FFF (one-sample)* | 63.60 | 75.04 | 62.62 | 69.56 | 65.26 | 49.05 | 66.29 | 63.28 |
| *FG-UAP* | 61.72 | 74.44 | 59.29 | 67.41 | 16.03 | 46.17 | 59.18 | 61.86 |
| *GD-UAP (no-data)* | 63.56 | 75.42 | 62.50 | 69.76 | 74.12 | 49.35 | 65.83 | 63.56 |
| *GD-UAP (mean-std)* | 63.39 | 74.85 | 62.17 | 69.71 | 75.55 | 50.35 | 65.69 | 63.61 |
| *GD-UAP (one-sample)* | 63.21 | 74.97 | 62.33 | 69.74 | 70.04 | 49.38 | 66.01 | 63.54 |
| *L4A-base* | 63.25 | 75.37 | 62.80 | 70.13 | 21.30 | 49.76 | 64.45 | 63.17 |
| *L4A-fuse* | 63.09 | 75.67 | 63.00 | 70.34 | 22.42 | 49.92 | 64.45 | 63.14 |
| *L4A-ugs* | 63.57 | 75.54 | 63.34 | 70.17 | 78.73 | 48.83 | 66.13 | 63.63 |
| *PD-UAP* | 63.29 | 75.29 | 62.38 | 70.37 | 77.57 | 49.38 | 66.02 | 63.14 |
| *SSP* | 63.11 | 74.88 | 61.78 | 69.45 | 56.88 | 46.94 | 64.56 | 62.78 |
| *STD* | 62.80 | 75.38 | 62.16 | 69.12 | 71.44 | 50.05 | 65.91 | 63.02 |
| *UAP (DeepFooç)* | 63.57 | 75.43 | 63.37 | 70.34 | 56.74 | 50.37 | 66.17 | 63.65 |
| *UAPEPGD* | 63.76 | 75.64 | 63.71 | 70.36 | 80.23 | 51.30 | 66.49 | 63.48 |
| Clean Accuracy | 89.13 | 89.08 | 89.15 | 90.77 | 83.23 | 92.06 | 87.47 | 89.13 |
| IAA Avg. | 45.87 ↓49% | 44.48↓50% | 39.48↓56% | 50.74↓44% | 37.75↓55% | 41.79↓55% | 36.73↓58% | 45.95↓48.4% |
| UAP Avg. | 63.22 ↓29% | 75.21↓16% | 62.43↓30% | 69.77↓23% | 61.16↓27% | 49.33↓46% | 65.30↓25% | 63.22↓29% |
| Adv Avg. | 54.03 ↓39% | 58.94↓34% | 50.28↓44% | 59.69↓34% | 48.77↓41% | 45.34↓51% | 50.18↓426% | 54.08↓39% |

