# OpenReview forum: "Adversarial Robustness of  Self-Supervised Learning in Vision"
_ICLR.cc/2025/Conference — ICLR 2025 Conference Withdrawn Submission_

### Official Review · Reviewer_pkQQ · 2024-10-31

**Soundness:** 2
**Presentation:** 3
**Contribution:** 2
**Rating:** 5
**Confidence:** 3

**Summary:**

The paper investigates the adversarial robustness of self-supervised learning (SSL) models for visual tasks, benchmarking seven SSL models and one supervised model across various tasks including ImageNet classification, transfer learning, segmentation, and detection. Results indicate that SSL models generally demonstrate superior robustness compared to supervised learning, especially in ImageNet classification and transfer learning tasks, though this robustness advantage is less pronounced in segmentation and detection. The study further explores the impacts of model architecture and training duration on robustness, noting that architectural choices and longer training durations can slightly improve robustness.

**Strengths:**

1. The paper presents a comprehensive evaluation of SSL models' adversarial robustness across different tasks, providing valuable insights into SSL models' performance relative to supervised models.
2. The investigation of architectural factors and training durations adds depth, highlighting how different configurations impact adversarial resilience.
3. The presentation and writing of this paper is good.

**Weaknesses:**

1. The statistical significance of certain robustness differences is not consistently addressed, particularly for tasks where SSL and supervised models show minimal variation.
2. Limited exploration of specific segmentation and detection setups may restrict generalizability, as adversarial impacts on these tasks can vary significantly with different configurations. So the insights are not too deep.

**Questions:**

For insights 2, 3, and 4 described in the introduction section, can the authors provide a more in-depth analysis of the root cause? Otherwise, it is more like a pure measurement technical report.

---

### Official Review · Reviewer_x2WD · 2024-10-31

**Soundness:** 3
**Presentation:** 2
**Contribution:** 1
**Rating:** 3
**Confidence:** 4

**Summary:**

The author investigated the adversarial robustness of different SSL methods and compared them with supervised learning methods. The author provided experimental results, some of which offer insights that differ from previous research. The article found that SSL exhibits stronger robustness than supervised learning in certain cases.

**Strengths:**

1. The author researched a large number of SSL algorithms, which required considerable effort and time.
2. The author created exquisite visualizations of the data.
3. The authors tested robustness across various downstream tasks.

**Weaknesses:**

1. In Figure 2, the abbreviation is more precisely referred to as MoCo v3.
2. Since the article has already mentioned the impact of model architecture on robustness, why are only the results of ResNet50 provided?
3. The article expands its research focus to SSL; why did it only investigate joint-embedding methods? The methods based on mask modeling and generative models were missing.
4. As a review article, the existing experimental data is still insufficient.
5. The article lacks in-depth analysis and could at least provide some visualizations.
6. Lacks discussion on natural corruption. [1,2]

[1] Chhipa, P. C., Holmgren, J. R., De, K., Saini, R., & Liwicki, M. (2023). Can self-supervised representation learning Methods Withstand distribution shifts and corruptions? Proceedings of the IEEE/CVF international conference on computer vision (ICCV) workshops, 4467–4476.
[2] Zhong, Y., Tang, H., Chen, J., Peng, J., & Wang, Y.-X. (2022). Is Self-Supervised Learning More Robust Than Supervised Learning? (No. arXiv:2206.05259). arXiv. https://doi.org/10.48550/arXiv.2206.05259

**Questions:**

The article appears to focus solely on previous studies about SSL robustness, overlooking reviews of a similar nature. Could the authors briefly discuss the differences between this article and other review articles?
What are the key findings of this paper? Could the author provide a brief description?

---

### Official Review · Reviewer_sVXC · 2024-10-31

**Soundness:** 3
**Presentation:** 2
**Contribution:** 3
**Rating:** 5
**Confidence:** 5

**Summary:**

The paper explores the adversarial robustness of self-supervised learning models, comparing them with supervised learning models across a set of tasks including classification, transfer learning, segmentation, and detection. Specifically, it investigates seven SSL models, including Barlow Twins, BYOL, DINO, MoCoV3, SimCLR, SwAV, and VICReg, alongside a supervised ResNet50 model as a baseline. This study aims to understand how SSL impacts adversarial robustness and explores the effects of architectural choices, training duration, and attack types on model performance.

**Strengths:**

- The paper evaluates seven SSL models and a supervised model on a range of tasks, including classification, transfer learning, segmentation, and detection. This wide coverage provides a good assessment of adversarial robustness, presenting a broad view of SSL performance across different vision tasks.

- The authors employed a diverse set of adversarial attack types including both instance adversarial attacks and universal adversarial perturbations. This level of detail is helpful for understanding which SSL methods are more robust against generalized attacks and which struggle under specific attack conditions.

- The paper highlights that SSL robustness doesn’t carry over equally to all tasks. For example, some SSL models may not be as robust in tasks like segmentation compared to classification. This helps users know when SSL might be less reliable in handling adversarial attacks.

- The authors show that contrastive SSL models like MoCoV3 can be just as robust as non-contrastive models in some cases, which challenges the common belief that non-contrastive methods are always better. This gives a more balanced view for choosing SSL models based on specific needs.

**Weaknesses:**

- For detection and segmentation tasks, the standard approach is to train models from scratch (end-to-end training with pretrained backbone) rather than freezing the backbone, as freezing is neither typical nor particularly beneficial. When training end-to-end, the weights change significantly, meaning there is no clear relationship between the pretraining algorithm and downstream robustness. Consequently, because the weights are altered during training, the authors cannot attribute any observed robustness solely to the pretraining algorithm.

- The study’s focus on just ResNet and a limited use of ViTs doesn’t fully explore how different architectures affect robustness. Testing more models (different sizes) and training for varied durations would give a clearer picture of SSL’s robustness potential.

- Only white-box attacks are considered, which limits the robustness evaluation. Including black-box and grey-box attacks would provide a more complete understanding of SSL robustness across different adversarial scenarios.

- For transfer learning, the study relies only on linear evaluation, which may miss some of SSL’s robustness benefits. Since real-world scenarios often involve fine-tuning, this limits the broader applicability of the results.

- While the authors mention that data augmentation could play a role in robustness, they don’t explore it deeply. Testing different augmentation strategies could reveal useful insights for improving SSL robustness against adversarial attacks.

- Key SSL models like MAE and DINOv2 were not included. Their inclusion would offer a more comprehensive assessment of SSL model robustness.

- The finding that SSL models are generally more robust than supervised ones is not new, as this has been explored in prior research [1].

- The COCO dataset, the standard benchmark for object detection, is missing. Including it would improve relevance and allow for better comparison with other studies.

- There are limited details on attack objectives for the detection and segmentation tasks, such as object vanishing, fabrication, or mislabeling. Since different objectives could impact robustness, more detail here would clarify the nature of the attacks used.

- While possibly beyond the main scope, evaluating models against poisoning and backdoor attacks would strengthen the paper’s general claims about SSL robustness. These attack types are increasingly relevant for assessing real-world model security.

- The source code is not provided, making it difficult to reproduce results or conduct further experiments. Providing code would benefit practitioners and researchers who want to build on this work.

[1] Goldblum, Micah, et al. "Battle of the backbones: A large-scale comparison of pretrained models across computer vision tasks." Advances in Neural Information Processing Systems 36 (2024).

**Questions:**

1. In Table 2, is the epsilon value set to 1/255 or 1/1, and what step size is used for the PGD attack?
2. Are there any results for patch attacks, and if not, was there a specific reason for their exclusion?
3. Can the authors clarify which attack objectives were prioritized (e.g., object vanishing, fabrication, or mislabeling), as different objectives might affect model robustness?
4. Was there a particular reason the COCO dataset was not used for detection, given its standard role in benchmarking?
5. Are there any insights on how hyperparameter choices (e.g., learning rate, batch size) might impact the adversarial robustness of SSL models in this study?

---

### Note · Authors · 2024-11-23

I have read and agree with the venue's withdrawal policy on behalf of myself and my co-authors.